# Lyapunov Guidance: Stabilizing Generative Flows with One-Line Code

## Abstract

Flow matching has recently emerged as a powerful approach to learning complex data distributions with excellent performance across diverse generative tasks, yet adapting pre-trained flow models to new tasks typically requires costly retraining. To mitigate this issue, post-training guidance methods were proposed as they are lightweight and user-friendly for downstream applications. However, existing guidance methods are unreliable since they usually rely on function approximations and lack structural guarantees of sampling stability. In this paper, we address this challenge by proposing a unified framework, LyaGuide (Lyapunov Guidance for flow matching), which reformulates the guidance in flow matching as a Lyapunov control problem. LyaGuide supports two modes depending on whether the Lyapunov function is a known priori: a model-driven mode for developer-oriented scenarios where the guidance distribution is explicitly specified, and a data-driven mode for user-oriented scenarios where pre-trained models can be adapted with downstream task-specific data. Furthermore, to enforce the stability, we introduce a pseudo projection operator with a closed-form expression that strictly satisfies the Lyapunov condition. Notably, LyaGuide is compatible with any guidance method and can be implemented with a single line of code. Experiments on synthetic datasets, image inverse problems, RL planning and EBM tasks demonstrate that our framework consistently improves sample quality and guidance fidelity while preserving efficiency, and it significantly enhances the performance of existing guidance methods.

## 1 Introduction

Generative modeling has recently witnessed remarkable progress with the advent of diffusion models (Song et al., 2021; Dhariwal & Nichol, 2021; Ho & Salimans) as well as their deterministic counterparts based on flow matching (Lipman et al., 2023; Liu et al., 2023b; Tong et al., 2023). Flow matching learns a time-dependent vector field that transfers an easy-to-sample base distribution to a complex data distribution, providing a mathematically elegant and computationally efficient alternative compared to stochastic diffusion processes. Nevertheless, it is still hard to adapt a pre-trained flow model to achieve fantastic performance in downstream applications by retraining with corresponding task-specific data, which is straightforward but suffers from a heavy and inflexible workflow. In contrast, post-training guidance methods are more lightweight and efficient for adaption, while existing approaches are almost heuristic or approximation-based Fan et al. (2025); Feng et al. (2025), lacking theoretical guarantees on the stability of the guided generation process. This urgently calls for a unified theoretical framework that unifies diverse conditions, including class labels (Dhariwal & Nichol, 2021; Ho & Salimans), structural constraints in molecular design (Zhang et al., 2024b), and reward functions in reinforcement learning (Jan-

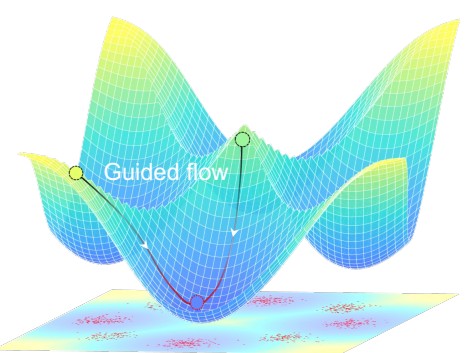

Figure 1: Lyapunov landscape for 8-Gaussian guidance from initial (black dashed) to target data (red dashed), defined as the negative of the energy landscape.

ner et al., 2022), while providing both improved performance and rigorous stability guarantees for guided flows.

An analogous challenge has long been studied in cybernetics, where the central objective is to steer the dynamics to the target states. Previous works employ Lyapunov stability theory to design stabilizing policies for linear or polynomial systems, such as the linear quadratic regulator (LQR) (Khalil, 2002) and the semidefinite programming (SDP) based sum-of-squares (SOS) methods (Parrilo, 2000). For more intricate high-dimensional and nonlinear dynamics, recent work has integrated machine learning into control (Tsukamoto et al., 2021). In particular, neural controllers are trained jointly with certificate functions, such as Lyapunov functions, LaSalle's invariants, barrier certificates, and contraction metrics (Chang et al., 2019; Zhang et al., 2022a; Yang et al., 2025; Zhang et al., 2022b; Qin et al., 2020; Sun et al., 2021).

In this work, we propose a unified flow matching guidance framework from the perspective of *Lyapunov control theory* (Artstein, 1983; Sontag, 1989; Polyakov, 2012), where the energy function that determines the conditional distribution is interpreted as a Lyapunov function, while the added guidance term serves as a stabilizing control input. This equivalence allows us to reinterpret a wide range of existing guidance strategies as special cases of Lyapunov control, as shown in Fig. 1, including classifier guidance, classifier-free guidance, energy-based guidance, and reward-guided generation. In addition to providing theoretical clarity, we also introduce *pesudo projection operation* that enforces the Lyapunov condition on the guidance, ensuring that the guided flows could converge to the target distribution (see Section 4). Thus, our approach unifies diverse guidance strategies within a Lyapunov framework, providing both theoretical guarantees and practical efficiency.

Although the control-theoretic view offers strong guarantees, a practical challenge is how to obtain the Lyapunov function $V$ that encodes prior knowledge. Sometimes $V$ can be written explicitly (for example, classifier guidance with $V(\boldsymbol{x}) = -\log p(y|\boldsymbol{x})$ in image generation (Dhariwal & Nichol, 2021) or structural energy terms in protein design (Zhang et al., 2024b)), while in many other applications prior knowledge is only implicitly available through few-shot supervised learning Janner et al. (2022); Lee et al. (2025); Black et al. (2023). To address this, we propose to *learn Lyapunov functions from limited data of downstream tasks*, and then design guidance policies based on the learned Lyapunov function.

**Contribution.** The principal contributions of this work can be summarized as follows:

- We establish a unified theoretical framework that unifies diverse guidance strategies in flow matching as instances of Lyapunov control, thereby providing a common foundation for guided generative modeling.
- We propose a pseudo projection operator with a closed-form expression that enforces Lyapunov stability for any candidate guidance function, achieving rigorous guidance with a single-line code implementation.
- Building on our theory, we introduce **LyaGuide**, which is adaptive to different scenarios and offers several efficient variants. These variants enable users to trade off between flexibility and strictness of stability guarantees depending on the downstream task.
- We validate the proposed framework on synthetic datasets and image inverse problems, where it significantly improves the sample quality of baselines while preserving computational efficiency. The code for reproducing the results is available at `anonymous/LyaGuide`.

Due to the limit of page, we summarise the **Related Work** in Appendix A.3.

## 2 BACKGROUND

### 2.1 FLOW MATCHING

Flow matching (Lipman et al., 2023) is a ODE-based training framework for generative modeling, which learns continuous-time dynamics that transforms a simple prior distribution to a complex data distribution. Formally, let $p_0$ and $p_1$ denote the source and target distributions over $\mathbb{R}^d$, respectively. flow matching aims to learn a time-dependent vector field $\boldsymbol{u}(\boldsymbol{x}, t) \triangleq \boldsymbol{u}_t(\boldsymbol{x}) : \mathbb{R}^d \times [0, 1] \to \mathbb{R}^d$ that pushes $p_0$ to $p_1$ along a continuous path $\{p_t\}_{t \in [0,1]}$, such that the flow $\phi_t$ governed by the ODE:

$$\frac{d\phi_t(\boldsymbol{x})}{dt} = \boldsymbol{u}_t(\phi_t(\boldsymbol{x})), \quad \phi_0(\boldsymbol{x}) = \boldsymbol{x}, \tag{1}$$

transfers $\boldsymbol{x} \sim p_0$ to a sample $\phi_1(\boldsymbol{x}) \sim p_1$.

To learns a neural field $\hat{\boldsymbol{u}}_{\boldsymbol{\theta}}(\boldsymbol{x}, t)$ that matches the continuous path $\{p_t\}_{t \in [0,1]}$, a convenient way to define targets is via a latent variable $z \sim p(z)$ that indexes conditional bridges $p_t(\boldsymbol{x}_t \mid z)$ together with conditional vector fields $\boldsymbol{u}_{t|z}(\boldsymbol{x}_t \mid z)$. This induces the marginal path and vector field $p_t(\boldsymbol{x}_t) = \int p_t(\boldsymbol{x}_t \mid z) \, p(z) \, dz$, $\boldsymbol{u}_t(\boldsymbol{x}_t) = \int \boldsymbol{u}_{t|z}(\boldsymbol{x}_t \mid z) \, p(z \mid \boldsymbol{x}_t) \, dz$, and it is known that $\boldsymbol{u}_t$ generates the marginal path $p_t$ (see (Lipman et al., 2023)). To avoid the intractable term $p(z \mid \boldsymbol{x}_t)$ in $\boldsymbol{u}_t(\boldsymbol{x}_t)$, conditional flow matching (CFM) propose to train the model with an equivalent and tractable conditional objective (Lipman et al., 2023; Tong et al., 2023):

$$\mathcal{L}_{\text{cond}}(\boldsymbol{\theta}) = \mathbb{E}_{t \sim \mathcal{U}(0,1), \, z \sim p(z), \, \boldsymbol{x}_t \sim p_t(\boldsymbol{x}_t|z)} \Big[ \big\| \hat{\boldsymbol{u}}_{\boldsymbol{\theta}}(\boldsymbol{x}_t, t) - \boldsymbol{u}_{t|z}(\boldsymbol{x}_t \mid z) \big\|_2^2 \Big],$$

whose minimizer coincides with that of conditional loss while remaining simulation-free and easy to estimate.

## 2.2 LYAPUNOV CONTROL THEORY

To begin with, we consider the feedback-controlled dynamic system of the following general form:

$$\dot{\boldsymbol{x}} = \boldsymbol{f}_t(\boldsymbol{x}, \boldsymbol{u}(\boldsymbol{x})), \; \boldsymbol{x} \in \mathbb{R}^d, \; \boldsymbol{u} \in \mathbb{R}^m, \tag{2}$$

where $\boldsymbol{f}_t$ is the Lipschitz-continuous vector field acting on some prescribed open set $\boldsymbol{x} \in \mathcal{D} \subset \mathbb{R}^d$. The solution initiated at time $t_0$ from $\boldsymbol{x}_0$ is denoted by $\boldsymbol{x}_t(t_0, \boldsymbol{x}_0)$. We assume that the stationary target position of the controlled system is the origin, i.e. $\boldsymbol{f}_t(\boldsymbol{0}, \boldsymbol{0}) = \boldsymbol{0}$. One major problem in cybernetics field is to design stabilizing controller $\boldsymbol{u}(\boldsymbol{x})$ (Wiener, 2019) such that $\lim_{t \to \infty} \boldsymbol{x}_t(t_0, \boldsymbol{x}_0) = \boldsymbol{0}$, for any initial value $\boldsymbol{x}_0 \in \mathcal{D}$.

**Theorem 2.1** *(Mao, 2007) Suppose that there exists a continuously differentiable function $V : \mathcal{D} \to R$ that satisfies the following conditions:* (i) $V(\boldsymbol{0}) = 0$, (ii) $V(\boldsymbol{x}) \geq c\|\boldsymbol{x}\|^p$ *for some constants* $c, p > 0$, (iii) *and* $\mathcal{L}_{\boldsymbol{f}} V < -\delta V$, *for some* $\delta > 0$. [1] *Then, the system is exponentially stable at the origin, that is,* $\limsup_{t \to \infty} \frac{1}{t} \log \|\boldsymbol{x}(t; t_0, \boldsymbol{x}_0)\| \leq -\frac{\delta}{p}$. *Here $V$ is called a Lyapunov function.*

For a given dynamic system equation 2, the design of appropriate control policies that satisfies the Lyapunov condition in Theorem 2.1 has been a central topic (Chang et al., 2019; Zhang et al., 2022a; Dawson et al., 2023; Yang et al., 2025).

**Problem Statement.** We assume that flow matching has already learned a base vector field $\boldsymbol{u}$ transferring the noise distribution $p_0$ to the data distribution $p_1$. For downstream tasks, this vector field must be adapted to generate task-specific conditional distributions $\frac{1}{Z} p_1(\boldsymbol{x}) e^{-J(\boldsymbol{x})}$, which is achieved by introducing a Lyapunov function $V$ that relates to energy distribution $e^{-J}$ and encodes task-related priors, and thus the guided field $\boldsymbol{u} + \boldsymbol{c}$ rigorously samples from the target distribution, where $\boldsymbol{c}$ is a guidance control term to be designed such that it encodes the information of the Lyapunov function $V$, i.e., satisfies the Lyapunov condition.

There are mainly two scenarios depending on how prior knowledge is provided, which are described in detail as follows.

> **Scenario 1 (Developers-Oriented).** Domain knowledge can be explicitly formulated as an analytical potential $V(\boldsymbol{x})$, making the conditioning objective transparent.

Typical examples include classifier or classifier-free guidance with class labels (Dhariwal & Nichol, 2021; Ho & Salimans), structural constraints in protein design (Zhang et al., 2024b), and reward functions in reinforcement learning (Janner et al., 2022). Despite their different forms, we provide a unified framework to equate flow matching guidance in Scenario 1 with the Lyapunov control problem (see Section 5.1), and further introduce an efficient and training-free guidance policy grounded in this theoretical framework (see Section 4).

---

[1] $\mathcal{L}_{\boldsymbol{f}} V$ represent the Lie derivative of $V$ along the direction $\boldsymbol{f}$, i.e., $\mathcal{L}_{\boldsymbol{f}} V = \nabla V \cdot \boldsymbol{f}_t$.

**Scenario 2 (Users-Oriented).** Prior knowledge is implicit and can only be learned through few-shot learning with some data–score pairs $\{(\boldsymbol{x}_i, V_i)\}_{i=1}^n$, without an explicit conditioner.

In contrast to Scenario 1, the challenge here is to infer a Lyapunov function and corresponding control from sparse or noisy supervision. We address this problem in Section 5.2.

# 3    A Unified Lyapunov Guidance Framework for Flow Matching

To incorporate diverse forms of prior knowledge into the inference stage of flow matching, we propose a unified Lyapunov guidance framework that interprets guidance as control policy derived from Lyapunov theory. Specifically, we consider the guided vector field as $\boldsymbol{u}_t + \boldsymbol{c}_t$, where $\boldsymbol{u}_t$ is the learned base transport field from noise distribution to data distribution, and $\boldsymbol{c}_t$ is an auxiliary control term that steers the dynamics toward a conditional target distribution. To align with the common Lyapunov condition, we first introduce *local Lyapunov condition* that formalizes how the controlled dynamics can converge to a desired conditional mode with diverse high-density regions, i.e., attractors. We then propose the first main theorem showing that guided flow matching under unlimited time is equivalent to a Lyapunov control system, with the energy function acting as a Lyapunov function. This equivalence provides a rigorous justification for viewing guidance in generative modeling through the lens of stability theory.

Inspired by Mao (2007), we extend the traditional Lyapunov condition to the following local Lyapunov condition.

**Proposition 3.1** *(Local Lyapunov condition) For the controlled dynamics $\dot{\boldsymbol{x}} = \boldsymbol{u}_t(\boldsymbol{x}) + \boldsymbol{c}_t(\boldsymbol{x})$ under controller $\boldsymbol{c}_t(\boldsymbol{x}_t)$, suppose there exists a a continuous differentiable function $V : \mathcal{D} \rightarrow R$ that satisfies the following conditions:* (i) *for each local minimum point $\boldsymbol{x}^*$ of $V$, let $V_{\boldsymbol{x}^*}(\boldsymbol{x}) = V(\boldsymbol{x}) - V(\boldsymbol{x}^*)$, there exists a $\varepsilon$-neighborhood $O(\boldsymbol{x}^*, \varepsilon)$ such that $V_{\boldsymbol{x}^*}(\boldsymbol{x}) \geq c\|\boldsymbol{x} - \boldsymbol{x}^*\|^p$, $\forall \boldsymbol{x} \in O(\boldsymbol{x}^*, \varepsilon)$ for some constants $c, p > 0$,* (ii) *and $\nabla V(\boldsymbol{x}) \cdot (\boldsymbol{u}_t(\boldsymbol{x}) + \boldsymbol{c}_t(\boldsymbol{x})) < -\delta V$, $\forall \boldsymbol{x} \in O(\boldsymbol{x}^*, \varepsilon)$, for some $\delta > 0$. Then, the system is locally exponentially stable at $\boldsymbol{x}^*$, that is, $\|\boldsymbol{x}_t - \boldsymbol{x}^*\| \leq \|\boldsymbol{x}_0\| \mathrm{e}^{-\frac{\delta}{p} t}$. Here $V$ is called a Lyapunov function.*

The above local condition enables us to view the guidance process in flow matching as the synthesis of a stabilizing controller that enforces local convergence toward high-density regions of a target conditional distribution. With this formulation, the energy function commonly used in conditional generative modeling plays the role of a Lyapunov function that shapes the convergence geometry. Our main theorem is presented as follows.

**Theorem 3.2 (Equivalence between Guided Flow Matching and Lyapunov Control)** *For the flow model $\boldsymbol{u}_t(\boldsymbol{x}_t)$ that generates the probability path $p_t(\boldsymbol{x})$, finding the guidance $\boldsymbol{c}_t(\boldsymbol{x}_t)$ to the vector field $\boldsymbol{u}_t(\boldsymbol{x}_t)$ to perform conditional sampling $p_t'(\boldsymbol{x}) = \frac{1}{Z_t} p_t(\boldsymbol{x}) \mathrm{e}^{-J(\boldsymbol{x})}$ is equivalent to finding the controller that satisfies the local Lyapunov condition, where energy function $J(\boldsymbol{x})$ contributes to Lyapunov function as $V \propto (J + c)$ for some constant $c$, e.g., $V = J$. More specifically, here the equivalence refers to that there exists a control that is both locally Lyapunov stable and generates the guided probability path, which therefore accelerates the sampling process of guided flows.*

The proof is provided in Appendix A.2.1. Theorem 3.2 establishes a general equivalence between conditional generation in flow matching and Lyapunov-based control, where the guidance term acts as a stabilizing controller and the energy (or potential) function plays the role of a Lyapunov function. This theoretical result provides a unified perspective to re-interpret various existing guidance methods in generative modeling as instances of Lyapunov control.

To illustrate the wide applicability of the framework, we identify several representative guidance paradigms where the associated energy function and the conditional distribution naturally serve as Lyapunov functions, so that the guidance terms can be interpreted as controllers that minimize $V(\boldsymbol{x})$ during generation.

**Proposition 3.3** *The following commonly used guidance strategies in generative modeling can all be interpreted as Lyapunov control within our unified framework:*

- **Classifier Guidance**: *Given a trained classifier $p(\boldsymbol{y}|\boldsymbol{x})$, the Lyapunov function for guided distribution $p(\boldsymbol{x}|\boldsymbol{y})$ specified on conditioner $\boldsymbol{y}$ is $V_{\boldsymbol{y}}(\boldsymbol{x}) = \log p(\boldsymbol{y}|\boldsymbol{x})$.*
- **Reward Guidance**: *In reinforcement learning tasks with reward function $R(\boldsymbol{x})$, the Lyapunov function for guided distribution $\frac{1}{Z}p_t(\boldsymbol{x})e^{R(\boldsymbol{x})}$ concentrates probability mass in high-reward regions is $V(\boldsymbol{x}) = -R(\boldsymbol{x})$.*
- **Energy-Based Model (EBM) Guidance**: *For a target EBM $p(\boldsymbol{x}) \propto e^{-E(\boldsymbol{x})}$, the Lyapunov function is naturally $V(\boldsymbol{x}) = -E(\boldsymbol{x})$.*
- **Image inverse problems.** *Let the forward operator be $\boldsymbol{y} = H(\boldsymbol{x}) + \varepsilon$ with $\varepsilon \sim \mathcal{N}(0, \sigma^2 I)$. Then $p(\boldsymbol{y}|\boldsymbol{x}) \propto \exp\big(-\frac{1}{2\sigma^2}\|H(\boldsymbol{x}) - \boldsymbol{y}\|_2^2\big)$ and a natural Lyapunov function is $V_{\boldsymbol{y}}(\boldsymbol{x}) = \frac{1}{2\sigma^2}\|H(\boldsymbol{x}) - \boldsymbol{y}\|_2^2$, yielding guided sampling $p_t'(\boldsymbol{x}) \propto p_t(\boldsymbol{x}) \exp\big(-V_{\boldsymbol{y}}(\boldsymbol{x})\big)$ that enforces data consistency Song et al. (2023).*

In Appendix A.2.2, we provide proof of the proposition and further discuss the relationship between exitsing guidance methods and our framework.

**Variants of LyaGuide.** The Proposition 3.1 establishes exponential stability of the controlled system, so we denote the corresponding method as **LyaGuide-ES**. Under a weaker condition that $\delta = 0$ in Theorem 3.2, the equilibrium remains stable in the sense of asymptotic stability, i.e., $\lim_{t\to\infty}\|\boldsymbol{x}_t - \boldsymbol{x}^*\| = 0$. In this case, $V$ is also a valid Lyapunov function. In addition, we can also modify the Lyapunov condition for a stronger exponential convergence. Therefore, two natural variants of LyaGuide arise by either relaxing or strengthening the Lyapunov condition:

- **LyaGuide-AS (Asymptotic Stability).** Setting $\delta = 0$ reduces the condition to $\nabla V(\boldsymbol{x}) \cdot (\boldsymbol{u}_t(\boldsymbol{x}) + \boldsymbol{c}_t(\boldsymbol{x})) < 0$, which guarantees asymptotic convergence $\lim_{t\to\infty}\|\boldsymbol{x}_t - \boldsymbol{x}^*\| = 0$. This yields a weaker but broadly applicable guidance mechanism.
- **LyaGuide-CS (Component-wise Stability).** A stronger requirement is that each component marked by subscript $i$ satisfies $\nabla V(\boldsymbol{x})_i(\boldsymbol{u}_t(\boldsymbol{x}) + \boldsymbol{c}_t(\boldsymbol{x}))_i \leq -\delta V(\boldsymbol{x})$. This enforces descent along all directions and provides stricter stability guarantees for the guided flow.

These two variants offer flexible trade-offs between stability and practical applicability. Empirically, LyaGuide-ES and LyaGuide-CS demonstrate superior performance and greater robustness across tasks, and we therefore recommend them as the default variants in practice (see Appendix A.5).

# 4 PSEUDO PROJECTION OPERATION FOR LYAPUNOV GUARANTEE

Given the equivalence between the guided flow matching and Lyapunov control, we show how we can better design the guidance term from the perspective of Lyapunov principle. Before the introduction of Lyapunov guidance, we propose a novel pseudo projection method that efficiently corrects the guidance term learned by neural networks. As in neural control, the candidate guidance function in flow may not rigorously satisfy the Lyapunov condition across the entire state space, since it is often learned from finite data or heuristically constructed. To address this limitation, we propose a projection operator that enforces the Lyapunov condition by projecting any candidate guidance into the admissible set of Lyapunov-stable controls.

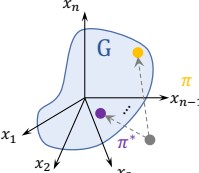

Figure 2: Illustration of the pseudo projection $\pi$ and exact projection $\pi^*$. Here the grey dot is the candidate control $\boldsymbol{c}$, the purple dot is the projected element $\pi^*(\boldsymbol{c}_t)$ of $\boldsymbol{c}$ in the target space $\mathcal{U}(V)$, and the yellow dot is the pseudo projected element $\pi(\boldsymbol{c}_t)$.

**Theorem 4.1 (Lyapunov Guarantee for Guidance)** *For a candidate controller $\boldsymbol{c}$ and the guidance controller space $\mathcal{U}(V) = \{\boldsymbol{u} : \nabla V \cdot (\boldsymbol{u}_t + \boldsymbol{c}_t) + \delta V \leq 0\}$ that rigorously satifies the local Lyapunov condition in Proposition 3.1, define the projection operator as*

$$\pi(\boldsymbol{c}_t, \mathcal{U}(V)) \triangleq \boldsymbol{c}_t - \frac{\max\big(0, \nabla V(\boldsymbol{x}) \cdot (\boldsymbol{u}_t(\boldsymbol{x}) + \boldsymbol{c}_t(\boldsymbol{x})) + \delta V(\boldsymbol{x})\big)}{\|\nabla V(\boldsymbol{x})\|^2} \nabla V(\boldsymbol{x}).$$

*Then $\pi(\boldsymbol{c}_t, \mathcal{U}(V))$ is locally Lipschitz continuous and thus the guided flow under $\pi(\boldsymbol{c}_t, \mathcal{U}(V))$ is well defined, and $\pi(\boldsymbol{c}_t, \mathcal{U}(V)) \in \mathcal{U}(V)$.*

The proof is provided in Appendix A.2.3. Fig. 2 shows the idea behind the pseudo projection, that any candidate guidance function can be systematically corrected via the pseudo projection operation, yielding a valid Lyapunov guidance that rigorously satisfies the stability condition across the state space. Therefore, our approach enjoys theoretical guarantees of conditional sampling via Lyapunov stability even when the initial candidate does not. Combining Theorem 3.2 with Theorem 4.1, we obtain a rigorous guarantee that flow model with projected guidance rigorously converges to the target distribution.

**Exact Projection vs. Pseudo Projection.** We refer to the operator in Theorem 4.1 as a *pseudo projection*, because for a feedback controller $c$, it is not the exact solution of the classical projection problem Deenen et al. (2021):

$$\pi^*(\boldsymbol{c}) \in \arg\min_{\tilde{\boldsymbol{c}}} \ \|\boldsymbol{c} - \tilde{\boldsymbol{c}}\|_{C(\mathbb{R}^d)},$$

$$\text{s.t.} \quad \nabla V(\boldsymbol{x}) \cdot (\boldsymbol{u}_t(\boldsymbol{x}) + \boldsymbol{c}(\boldsymbol{x})) \leq -\delta V, \quad \forall \boldsymbol{x} \in \mathcal{D}.$$

In general, obtaining a closed-form expression of the exact projection operator $\pi$ is intractable. A common alternative is to solve a simplified quadratic program (QP) at each state $\boldsymbol{x}$ Chow et al. (2019):

$$\tilde{\pi}(\boldsymbol{c})(\boldsymbol{x}) \in \arg\min_{\tilde{\boldsymbol{c}}(\boldsymbol{x})} \ \|\boldsymbol{c}(\boldsymbol{x}) - \tilde{\boldsymbol{c}}(\boldsymbol{x})\|,$$

$$\text{s.t.} \quad \nabla V(\boldsymbol{x}) \cdot (\boldsymbol{u}_t(\boldsymbol{x}) + \boldsymbol{c}(\boldsymbol{x})) \leq -\delta V(\boldsymbol{x}).$$

Although this guarantees the Lyapunov condition for each state $\boldsymbol{x}$, it requires solving an online optimization problem. In the context of flow matching, the dimensionality of the state space can be very high (e.g., images, molecular structures, or protein conformations). In such cases, solving a QP in every integration step would be prohibitively expensive. In contrast, the pseudo projection in Theorem 4.1 has an explicit analytical form, making it more efficient in practice while still ensuring that the guidance satisfies the Lyapunov condition.

In the context of flow matching, the dimensionality of the state space can be very high (e.g., images, molecular structures, or protein conformations). In such cases, solving a QP in each integration step would be prohibitively expensive. The pseudo projection offers a tractable alternative with closed-form updates, enabling stability enforcement during sampling without introducing substantial computational overhead. This makes the proposed approach particularly well suited for high-dimensional generative modeling tasks.

**Intuitive explanation of Theorems**. Our Theorem 3.2 shows that Lyapunov-stable controls and guidance-compatible controls are theoretically interchangeable, but the two directions differ greatly in difficulty. Converting a Lyapunov control into a guidance term requires solving auxiliary PDEs (see Appendix A.2.1), whereas converting a guidance term into a Lyapunov-stable one is much easier: a simple pointwise projection onto the Lyapunov-stable cone guarantees monotone decrease of $V$ while preserving the original guidance structure, as shown in Theorem 4.1 later. This is exactly the advantage exploited by LyaGuide—the projection step provides an efficient, PDE-free way to obtain a control that satisfies both stability and guidance properties.

## 5 LYAPUNOV GUIDANCE FOR DIFFERENT SCENARIOS

Building on the equivalence between guided flow matching and Lyapunov control that established in Section 3, we now specify the guidance policies for two different practical scenarios. The distinction lies in whether prior knowledge can be explicitly formulated: in Scenario 1, the Lyapunov function $V$ admits an analytical expression, and guidance can be synthesized directly (or learned from $V$); but in Scenario 2, the Lyapunov function is unknown and must be learned from data, which requires us to learn both $V$ and the associated control.

### 5.1 SCENARIO 1: EXPLICIT PRIOR KNOWLEDGE

When domain knowledge can be analytically represented, the Lyapunov function $V(\boldsymbol{x})$ is directly available. For such tasks, $V(\boldsymbol{x})$ acts as a certificate function that encodes domain-specific constraints, while the control term $\boldsymbol{c}(\boldsymbol{x}, t)$ is designed according to the Lyapunov condition in Proposition 3.1, ensuring that the guided flow is exponentially stable toward the high-density regions of the conditional distribution.

According to the projection operation described in Section 4, we begin with a candidate guidance function and then enforce the Lyapunov condition by projecting it onto the admissible function space. Within the flow matching literature, a common choice of candidate is based on the gradient of the Lyapunov function, $\boldsymbol{c}_t(\boldsymbol{x}) \propto -\nabla V(x)$, which is closely related to the score function of the prior energy distribution Song & Ermon (2019). The complete procedure is summarised in Algorithm 1.

## 5.2 SCENARIO 2: IMPLICIT PRIOR KNOWLEDGE VIA FEW-SHOT LEARNING

When explicit formulations of prior knowledge are unavailable, we assume access to a small set of preference data–score pairs $\{(\boldsymbol{x}_i, V_i)\}_{i=1}^n$, where $V_i$ represents a task-specific score or energy evaluated at $\boldsymbol{x}_i$. Our first goal is to recover a Lyapunov function $V_{\boldsymbol{\theta}_V}(\boldsymbol{x})$ from these pairs by minimizing a supervised regression loss: $\mathcal{L}_V(\boldsymbol{\theta}_V) = \frac{1}{n} \sum_{i=1}^n \left( V_{\boldsymbol{\theta}_V}(\boldsymbol{x}_i) - V_i \right)^2$.

**Local-minima-aware training of $V$.** According to Proposition 3.1, the Lyapunov function $V$ should correctly identify task-relevant local minima, i.e., low-energy and high-density regions where the Lyapunov guidance becomes active. To bias the learner toward these regions when only few-shot supervision $(\boldsymbol{x}_i, V_i)$ is available, we adopt a soft importance weighting on the targets,

$$w_i = \frac{\exp(-\alpha\,V_i)}{\sum_{j=1}^n \exp(-\alpha\,V_j)},$$

which places greater emphasis on low-$V$ samples. We then estimate $V_{\theta_V}$ via a weighted regression, optionally augmented with a smoothness regularizer, so that the learned potential captures the correct local minima structure needed for Lyapunov guidance.

$$\mathcal{L}_V(\boldsymbol{\theta}_V) = \frac{1}{n} \sum_{i=1}^n w_i \left( V_{\theta_V}(\boldsymbol{x}_i) - V_i \right)^2. \tag{3}$$

**Control design.** Once a valid Lyapunov function $V_{\boldsymbol{\theta}_V}$ is obtained, the corresponding guidance control can then be derived in two ways: either by explicit synthesis according to the Lyapunov framework,

$$\boldsymbol{c}(\boldsymbol{x}, t) = \arg \min_{\boldsymbol{c}} \ \nabla V_{\boldsymbol{\theta}_V}(\boldsymbol{x}) \cdot (\boldsymbol{u}_t(\boldsymbol{x}) + \boldsymbol{c}(\boldsymbol{x}, t)) + \delta V_{\boldsymbol{\theta}_V}(\boldsymbol{x}),$$

or by integrating $V_{\boldsymbol{\theta}_V}$ into existing guidance methods (e.g. classifier or reward guidance) to regularize their dynamics via a Lyapunov-inspired penalty. In this work we employ the training algorithm $g_\phi$ posed in Feng et al. (2025) for learning the guidance.

This two-stage design separates the estimation of the implicit energy landscape from the design of the guidance control, enabling user-provided supervision to be incorporated flexibly into flow matching. Although we present a two-stage procedure here (first learning $V$, then designing $\boldsymbol{c}$), in practice $V$ and $\boldsymbol{c}$ can also be optimized jointly under a Lyapunov-inspired loss, as applied in Zhang et al. (2022a; 2024a); Yang et al. (2025)

## 6 EXPERIMENTS

In this section, we demonstrate the superiority of the LyaGuide over existing methods with extensive experiments from low dimensional tasks to high dimensional tasks. We provide more details of the experiments and more experiments about EBM in Appendix A.5.

### 6.1 EVALUATION ON SYNTHETIC DATASET

**Scenario 1:** We first evaluate our approach on the synthetic datasets introduced in (Feng et al., 2025), where the source distribution $p_0$ is chosen as uniform (resp. circle) distribution and the target distribution $p_1$ is an 8-Gaussian mixture (resp. S-curve). For each dataset, we design a task-specific energy function $V$ to encode prior knowledge, as shown in Fig. 3 (first column on the left).

We compare against a wide range of existing guidance methods reported in Feng et al. (2025) as our baseline: contrastive energy guidance (CEG) Lu et al. (2023), Monte Carlo guidance ($g^{\text{MC}}$),

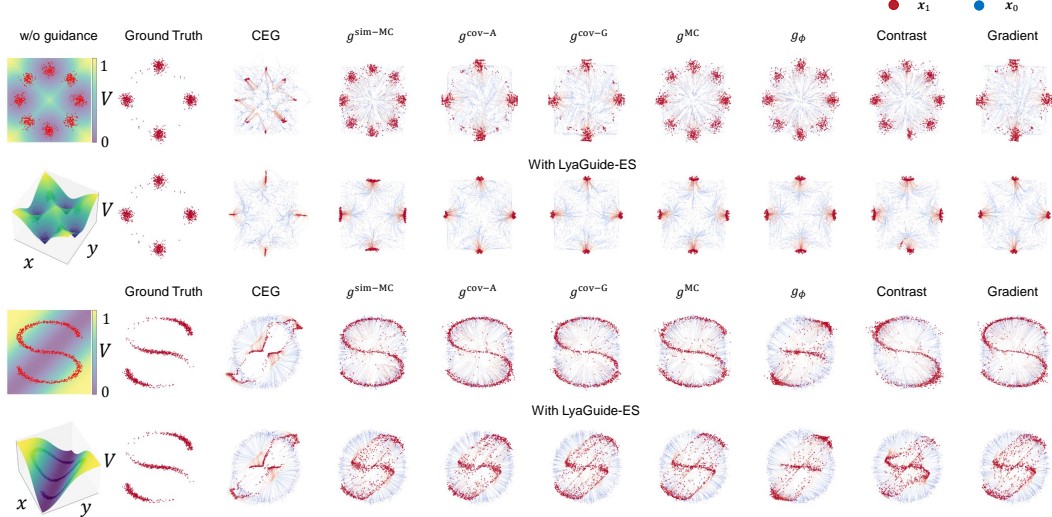

Figure 3: Scenario 1 results on synthetic dataset. For each target distribution, the top (resp. bottom) row correspond to the methods without (resp. with) LyaGuide. We visualize the start/end points and the flow trajectories. LyaGuide siginificantly improves the performance across different methods.

learned guidance ($g_\phi$), covariance-based approximations ($g^{\text{cov-A}}$ and $g^{\text{cov-G}}$), and simplified Monte Carlo ($g^{\text{sim-MC}}$). We also include contrastive guidance, which drives the flow toward unexpected regions, and gradient-based guidance commonly used in generative modeling Zhang et al. (2024b) for comparison. Each of these methods is first applied in its original form, and then we treat their guidance terms as candidates in Algorithm 1 and refine them with our proposed LyaGuide-ES. We also provide some detailed analysis results for LyaGuide-AS and LyaGuide-CS in Appendix A.5.

As shown in Fig. 3, all methods benefit from our Lyapunov guidance, where the guided trajectories become more stable and align more closely with the ground truth distribution. Specifically, for the first task (8-Gaussian mixture), we can see that most of the methods fail to sample correct trajectories without the guidance of LyaGuide, of which the best performance achieved by $g^{\text{cov-A}}$, $g^{\text{cov-G}}$ and *Gradient* is still defective, while noise at the four corners of the diagonals drops off drastically or even disappears after application of LyaGuide. Similar effect can be observed for the second task (S-curve distribution), where the LyaGuide optimized methods exhibit sharper mode coverage and fewer spurious samples compared to the original versions. These results demonstrate

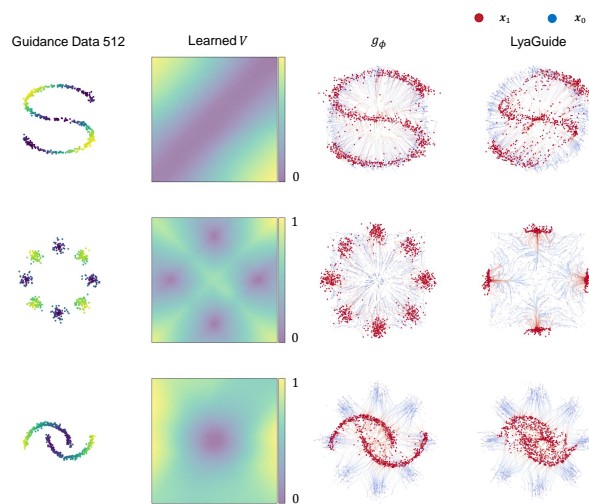

Figure 4: Scenario 2 results with dataset size = 512.

that LyaGuide provides an effective and practical way to improve guidance quality, consistently achieving gains across all baselines and energy functions.

**Boundary effects of Pseudo Projection.** We also note that there are two subtleties in the second task. First, CEG+Lya shows little difference with CEG, as the original samples already lie in the trough of $V$; since our projection encourages trajectories toward the trough, no further adjustment occurs. Second, Contrast+Lya does not preserve the exact masked S-curve shape. Instead, samples

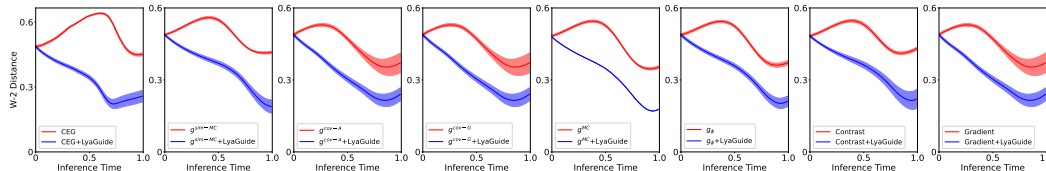

Figure 5: Inference-time vs. Wasserstein-2 distance on the 8-Gaussian mixture task (1024 test samples). Curves show the mean over five runs, and shaded regions denote standard deviations..

concentrate along the boundary of the trough of $V$, consistent with our projection principle: enforcing the Lyapunov inequality $\{\boldsymbol{x} : \nabla V(\boldsymbol{x}) \cdot (\boldsymbol{u}_t(\boldsymbol{x}) + \boldsymbol{c}_t(\boldsymbol{x})) \leq -\delta V(\boldsymbol{x})\}$. Thus, data near the peaks of $V$ are guided toward the nearest trough boundary, rather than the lower-left or upper-right arms of the S-curve.

**Scenario 2:** We next evaluate LyaGuide in Scenario 2, where explicit prior knowledge is not available and only a small set of preference data–score pairs is provided. Following the setup in Section 5.2, we learn a Lyapunov function $V_{\boldsymbol{\theta}_V}$ first and then derive the guidance control either through explicit synthesis or by integrating into existing methods. We consider synthetic benchmarks with three tasks, and vary the number of supervision pairs $\{(\boldsymbol{x}_i, V_i)\}$ from 128 to 1024.

As shown in Fig. 4, even with very sparse supervision, LyaGuide successfully recovers meaningful Lyapunov landscapes whose minima align with the high-density regions of the target distribution. Compared to the baseline $g_\phi$, the LyaGuide method enforces stability throughout the trajectory, yielding better guided flows that concentrate around the correct modes while suppressing spurious samples. We further perform the ablation study on data size in Appendix A.5.

**Acceleration of Inference Speed.** To evaluate how the sampling horizon of flow matching influences the behaviour of LyaGuide, we perform an early-inference-termination study on the 8-Gaussian mixture task. Instead of integrating the flow dynamics up to $t = 1$, we interrupt the evolution at intermediate times and compute the Wasserstein-2 distance between the current particle distribution and the target mixture. Figure 5 reports the mean and standard deviation over five independent trials. Across all eight guidance mechanisms, LyaGuide exhibits substantially faster convergence than their unguided counterparts. Even at early inference times (e.g., $t \approx 0.4$), the guided trajectories have already contracted much closer to the target distribution, while the baseline flows remain noticeably farther away. This indicates that the Lyapunov-structured correction not only improves final sample quality but also accelerates the transient approach to the target distribution. In addition, the performance gap remains robust across guidance types, demonstrating that LyaGuide consistently enhances the efficiency of flow-matching inference and reduces the reliance on long integration horizons.

**Ablation study** We further examine the influence of the scaling parameters $\delta$ and $k$ on the behaviour of LyaGuide. The results are summarised in Fig. A.5.1 and Table 3. The parameter $\delta$ affects both the Lyapunov convergence rate and the projection strength, making it a global hyperparameter. Empirically, LyaGuide is robust across a wide range of $\delta$, where larger values lead to stronger contraction toward low-energy regions but may reduce exploration. A moderate choice of $\in (0, 1]$ consistently provides a good balance between stability and diversity. The coefficient $k$ influences only gradient-based guidance methods by adjusting the strength of the initial guidance term $-k\nabla V$. Increasing $k$ generally improves the stability and quality of gradient-based guidance without affecting other methods. Overall, the method demonstrates low sensitivity to both parameters within reasonable ranges.

## 6.2 Image Inverse Problems

We further validate LyaGuide on high-dimensional image inverse problems, which have become a standard benchmark for assessing guidance quality in flow matching (Song et al., 2023; Feng et al., 2025). The objective is to reconstruct a clean image $\boldsymbol{x}$ from a corrupted observation $\boldsymbol{y} = H(\boldsymbol{x}) + \varepsilon$, where $H$ is a known degradation operator and $\varepsilon$ denotes Gaussian noise. In contrast to pseudoinverse-guided diffusion (Song et al., 2023), our projection-based scheme offers a lightweight alternative that strictly enforces stability during sampling.

We evaluate on the CelebA-HQ dataset under the box inpainting setting, where a central square region of each image is masked and the model must complete the missing content. Performance is measured by FID, LPIPS, PSNR, and SSIM on 3000 test samples. As summarized in Table 1, LyaGuide consistently improves all baseline guidance methods. Similar improvements are also observed on two additional inverse problems, namely super-resolution and Gaussian deblurring, with detailed comparisons reported in Appendix A.5.

Table 1: Image inverse problem results on CelebA-HQ (Box painting task). The best and runner-up results are highlighted with **bold** and underline, respectively.

| | | Original Methods | | | | LyaGuide-ES | | | | LyaGuide-CS | | | |
|---|---|---|---|---|---|---|---|---|---|---|---|---|---|
| | | FID ↓ | LPIPS ↓ | PSNR ↑ | SSIM ↑ | FID ↓ | LPIPS ↓ | PSNR ↑ | SSIM ↑ | FID ↓ | LPIPS ↓ | PSNR ↑ | SSIM ↑ |
| OT-CFM | $g^{\text{cov-A}}$ | 7.3387 | 0.1907 | 25.5984 | 0.8431 | 7.4039 | 0.1898 | 25.6935 | 0.8442 | **6.4722** | **0.1739** | 25.6217 | **0.8562** |
| | $g^{\text{sim-A}}$ | 19.8569 | 0.2309 | 26.4127 | 0.7950 | 19.8852 | 0.2309 | **26.4741** | 0.7950 | 12.7344 | 0.1921 | 26.3142 | 0.842 |
| | ΠGDM | 30.3839 | 0.3200 | 20.8253 | 0.7193 | 30.3839 | 0.3200 | 20.8253 | 0.7193 | 26.785 | 0.291 | 21.3654 | 0.7473 |
| | $g^{\text{MC}}$ | 18.6635 | 0.2391 | 26.9492 | 0.8124 | 24.1950 | 0.2396 | 26.9624 | 0.8129 | 16.0392 | 0.2283 | 27.0681 | 0.8200 |
| CFM | $g^{\text{cov-A}}$ | 7.6629 | 0.1922 | 25.8612 | 0.8414 | 7.6819 | 0.1918 | 25.9367 | 0.8419 | 6.9783 | 0.1764 | 25.8155 | **0.854** |
| | $g^{\text{sim-A}}$ | 9.7060 | 0.1935 | 26.0867 | 0.8263 | 9.6002 | 0.1935 | 26.1094 | 0.8263 | **6.8763** | **0.1629** | 26.0592 | 0.8508 |
| | $g^{\text{cov-G}}$ | 19.8022 | 0.2379 | 27.0087 | 0.8138 | 24.3271 | 0.2379 | 27.0017 | 0.8145 | 22.0999 | 0.2293 | **27.0301** | 0.8206 |
| | ΠGDM | 19.0847 | 0.2323 | 25.6418 | 0.8093 | 19.0619 | 0.2336 | 25.9002 | 0.8074 | 14.1461 | 0.2003 | 26.112 | 0.8439 |

## 6.3 LyaGuide on RL Planning

To further strengthen the empirical validity of our framework and enhance the sufficiency of benchmarks, we additionally evaluate LyaGuide on standard offline RL planning tasks in the D4RL Locomotion suite, following the experimental protocol used in prior work Feng et al. (2025). In this setting, generative models are employed as planners by sampling from distributions proportional to $\exp(R(x))$, where $R$ denotes the environment return. We directly compare each baseline guidance method and its LyaGuide-enhanced counterpart under both CFM and OT-CFM generative models, using the average normalized score over five runs as the evaluation metric.

The results are summarized in Table 2, and demonstrate a consistent performance improvement across nearly all tasks and guidance types. In particular, LyaGuide yields higher normalized returns for HalfCheetah, Hopper, and Walker2d across Medium-Expert, Medium, and Medium-Replay settings. Averaged over all tasks, LyaGuide improves the best-performing baseline by a clear margin under both CFM and OT-CFM. These findings confirm that LyaGuide enhances the contractive behavior of guidance dynamics in planner-based RL, leading to more reliable and higher-quality action-sequence generation.

Table 2: Results of the D4RL Locomotion experiments. The best results are highlighted with **bold**.

| | | | CFM | | | | | OT-CFM | | | | |
|---|---|---|---|---|---|---|---|---|---|---|---|---|
| | | | $g^{\text{cov-A}}$ | $g^{\text{cov-G}}$ | $g^{\text{sim-MC}}$ | $g^{\text{MC}}$ | $g_\phi$ | $g^{\text{cov-A}}$ | $g^{\text{cov-G}}$ | $g^{\text{sim-MC}}$ | $g^{\text{MC}}$ | $g_\phi$ |
| Without LyaGuide | Medium-Expert | HalfCheetah | 44.9 | 50.3 | 62.7 | 58.9 | 43.0 | 50.4 | 49.3 | 47.7 | 80.2 | 60.5 |
| | | Hopper | 88.9 | 84.3 | 96.9 | 1.04 | 84.7 | 92.6 | 111.1 | 74.9 | 108.4 | 75.5 |
| | | Walker2d | 64.5 | 95.7 | 83.7 | 86.2 | 78.0 | 76.4 | 75.8 | 94.4 | 102.5 | 56.5 |
| | Medium | HalfCheetah | 41.1 | 43.1 | 42.5 | 40.6 | 43.1 | 43.4 | 42.3 | 34.4 | 41.6 | 42.4 |
| | | Hopper | 67.5 | 83.6 | 73.6 | 71.8 | 69.2 | 70.6 | 67.3 | 63.7 | 70.9 | 66.1 |
| | | Walker2d | 75.5 | 68.5 | 74.7 | 78.2 | 50.8 | 75.4 | 77.4 | 77.3 | 78.7 | 67.6 |
| | Medium-Replay | HalfCheetah | 35.4 | 33.8 | 26.1 | 34.3 | 29.7 | 30.1 | 31.2 | 19.5 | 34.3 | 29.2 |
| | | Hopper | 44.7 | 50.3 | 46.4 | 56.2 | 44.9 | 48.4 | 57.5 | 49.9 | 63.4 | 50.3 |
| | | Walker2d | 47.5 | 48.1 | 47.4 | 53.5 | 38.6 | 0.5390 | 45.7 | 30.8 | 59.0 | 43.3 |
| | Average of Baselines | | 56.7 | 61.9 | 61.5 | 64.8 | 53.5 | 60.1 | 61.9 | 54.7 | 71.0 | 54.6 |
| Without LyaGuide | Medium-Expert | HalfCheetah | 56.7 | 65.6 | 58.8 | 88.0 | 58.5 | 50.6 | 61.6 | 57.9 | 88.6 | 64.2 |
| | | Hopper | 102.6 | 104.2 | 84.2 | 112.2 | 85.8 | 103.5 | 102.8 | 75.7 | 111.3 | 99.4 |
| | | Walker2d | 78.4 | 84.4 | 80.4 | 98.4 | 92.5 | 79.7 | 88.0 | 87.3 | 105.5 | 76.1 |
| | Medium | HalfCheetah | 42.3 | 42.32 | 43.7 | 43.0 | 42.6 | 40.6 | 41.8 | 42.0 | 44.3 | 43.1 |
| | | Hopper | 81.8 | 75.8 | 75.8 | 75.9 | 71.5 | 74.8 | 74.7 | 61.8 | 73.0 | 72.0 |
| | | Walker2d | 72.9 | 76.6 | 75.80 | 79.9 | 70.8 | 76.8 | 78.5 | 79.6 | 73.8 | |
| | Medium-Replay | HalfCheetah | 32.6 | 36.8 | 35.7 | 38.5 | 30.4 | 28.9 | 35.5 | 26.5 | 41.3 | 25.3 |
| | | Hopper | 58.2 | 64.1 | 54.4 | 59.5 | 53.5 | 60.8 | 59.6 | 59.5 | 73.4 | 47.6 |
| | | Walker2d | 51.3 | 64.9 | 45.0 | 65.6 | 54.7 | 62.1 | 53.4 | 52.2 | 69.0 | 43.0 |
| | Average of LyaGuide | | **64.1** | **68.3** | **61.5** | **73.5** | **63.2** | **63.5** | **66.0** | **60.5** | **76.2** | **60.5** |

## 7 Scope and limitations

**Finite-Time Lyapunov Control**. Our framework currently relies on asymptotic Lyapunov stability, which ensures convergence as $t \to \infty$ but does not provide accurate convergence time for Lyapunov

control. Since generative models operate with fixed inference time, a natural question is whether finite-time or prescribed-time Lyapunov control can be incorporated to regulate convergence within this horizon. While classical finite-time theory (Polyakov, 2012; Polyakov et al., 2015) guarantees bounded-time stability, its direct use may neglect the data distribution $p_1(\boldsymbol{x}) = q(\boldsymbol{x})$. Future work could therefore explore finite-time Lyapunov methods or design distribution-level Lyapunov functionals that couple $q(x)$ with the task-specific energy $J(x)$, enabling faithful generation with controlled convergence speed.

**On the Role of the Potential $V$.** Our guarantees hinge on the availability or learnability of a potential $V$ that meaningfully encodes task priors. When $V$ is poorly specified or underfitted, guidance may bias samples toward suboptimal regions, a limitation shared with classifier/EBM/reward guidance in diffusion (Dhariwal & Nichol, 2021; Ho & Salimans; Janner et al., 2022). Scenario 2 mitigates this by learning $V_\theta$ from few-shot supervision with a Lyapunov penalty, but the expressiveness–regularity trade-off remains: overly flexible $V_\theta$ may violate smoothness or curvature required for stabilization and overly rigid $V_\theta$ may underfit complex priors, how to address such issue in learning Lyapunov function is a future research direction.

## 8  CONCLUSION

In this work, we introduce LyaGuide, a unified framework that reformulates guidance in flow matching as a Lyapunov control problem. By establishing a theoretical equivalence between guided flows and Lyapunov stabilization control process, we show that diverse guidance strategies can be interpreted within a single control-theoretic perspective. To rigorously enforce stability, we also propose a pseudo projection operator with a closed-form expression, which guarantees that any candidate guidance satisfies the Lyapunov condition. Notably, this projection can be implemented with a single line of code and is compatible with existing guidance methods, consistently enhancing their performance. This work opens new opportunities for control-inspired generative modeling, particularly in domains where explicit priors are scarce but limited supervision can be leveraged, making the framework especially suitable for deploying pre-trained models on individual systems.

## REPRODUCIBILITY STATEMENT

The code of the experiments used in this paper is available at: `https://anonymous.4open.science/r/LyaGuide_ICLR2026-F3BB`, and complete proofs of our theoretical claims can be found in Appendix A.2.

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

# A APPENDIX

## A.1 ALGORITHMS

---

**Algorithm 1:** LyaGuide: Training-free Approach

---

1: **Input:** Learned vector field $u_t$ in flow matching, explicit Lyapunov function $V(x)$ for prior knowledge, guidance parameter $\delta$, initial parameter set $k$.

2: **Candidate guidance**: E.g., $\boldsymbol{c}_t(\boldsymbol{x}) = -k\nabla V(\boldsymbol{x})$      ▷ user-defined

3: **Pseudo projection operation** for rigorous satisfaction of Lyapunov condition 3.1:

4: **LyaGuide-ES:** $\boldsymbol{c}_t^*(\boldsymbol{x}) = \boldsymbol{c}_t(\boldsymbol{x}) - \frac{\max(0, \nabla V \cdot (\boldsymbol{u}_t + \boldsymbol{c}_t) + \delta V)}{\|\nabla V\|^2} \cdot \nabla V$

5: **LyaGuide-AS:** $\boldsymbol{c}_t^*(\boldsymbol{x}) = \boldsymbol{c}_t(\boldsymbol{x}) - \frac{\max(0, \nabla V \cdot (\boldsymbol{u}_t + \boldsymbol{c}_t))}{\|\nabla V\|^2} \cdot \nabla V$

6: **LyaGuide-CS:** $\boldsymbol{c}_t^*(\boldsymbol{x})_i = (\boldsymbol{c}_{\boldsymbol{\theta}_c}(\boldsymbol{x}, t))_i - \frac{\max(0, (\nabla V)_i(\boldsymbol{u} + \boldsymbol{c})_i + \delta V)}{\|\nabla V\|^2} \cdot \nabla V$

7: **Output:** Guided vector field $\boldsymbol{f}_t(x) = \boldsymbol{u}_t(\boldsymbol{x}) + \boldsymbol{c}_t^*(\boldsymbol{x})$ for conditional flow matching.

---

The score-based candidate guidance in Algorithm 1 provides an efficient approximation to the ideal guidance, although its performance depends on the choice of the initial parameter $k$. A better candidate may be obtained by iteratively running the algorithm with different values of $k$ and selecting the best outcome. More generally, any other method for generating a guidance term can be seamlessly integrated into our framework Feng et al. (2025). In particular, one may train a neural network to approximate the candidate guidance, as we demonstrate in Scenario 2 below.

---

**Algorithm 2:** LyaGuide: Training-based Approach

---

**Input:** Preference data-score pairs $\mathcal{D} = \{(\boldsymbol{x}_i, V_i)\}_{i=1}^n$, learning rate $\beta$, initial parameters $\boldsymbol{\theta}_0$, $\boldsymbol{\theta} = (\boldsymbol{\theta}_V, \boldsymbol{\theta}_c)$, learned flow $\boldsymbol{u}_t$ from take noisy distribution $p_0$ to data distribution $p_1$, guidance parameter $\delta$.

**while** *not converged* **do**

     $(\boldsymbol{x}_i, V_i) \sim \mathcal{D}$      ▷ sample data and score

     Compute Lyapunov Loss $L(\boldsymbol{\theta}_V)$ from equation 3

     $\boldsymbol{\theta}_V \leftarrow \boldsymbol{\theta}_V - \beta \nabla_{\boldsymbol{\theta}} L(\boldsymbol{\theta}_V)$      ▷ Learn Lyapunov function

     Train $\boldsymbol{\theta}_c$ with any existing learning-based guidance method

**Pseudo projection operation :**

**LyaGuide-ES:** $\boldsymbol{c}_t^*(\boldsymbol{x}) = \boldsymbol{c}_{\boldsymbol{\theta}_c}(\boldsymbol{x}, t) - \frac{\max(0, \nabla V_{\boldsymbol{\theta}_V} \cdot (\boldsymbol{u} + \boldsymbol{c}) + \delta V_{\boldsymbol{\theta}_V})}{\|\nabla V_{\boldsymbol{\theta}_V}\|^2} \cdot \nabla V_{\boldsymbol{\theta}_V}$

**LyaGuide-AS:** $\boldsymbol{c}_t^*(\boldsymbol{x}) = \boldsymbol{c}_{\boldsymbol{\theta}_c}(\boldsymbol{x}, t) - \frac{\max(0, \nabla V_{\boldsymbol{\theta}_V} \cdot (\boldsymbol{u} + \boldsymbol{c}))}{\|\nabla V_{\boldsymbol{\theta}_V}\|^2} \cdot \nabla V_{\boldsymbol{\theta}_V}$

**LyaGuide-CS:** $\boldsymbol{c}_t^*(\boldsymbol{x})_i = (\boldsymbol{c}_{\boldsymbol{\theta}_c}(\boldsymbol{x}, t))_i - \frac{\max(0, (\nabla V_{\boldsymbol{\theta}_V})_i(\boldsymbol{u} + \boldsymbol{c})_i + \delta V_{\boldsymbol{\theta}_V})}{\|\nabla V_{\boldsymbol{\theta}_V}\|^2} \cdot \nabla V_{\boldsymbol{\theta}_V}$

**Output:** Guided vector field $\boldsymbol{f}_t(x) = \boldsymbol{u}_t(\boldsymbol{x}) + \boldsymbol{c}_t^*(\boldsymbol{x})$ for conditional flow matching.

---

## A.2 THEORETICAL PROOFS

**Notations.** Denote by $\|\cdot\|$ the $L^2$-norm for any given vector in $\mathbb{R}^d$. Denote by $\|\cdot\|_{C(\mathcal{D})}$ the maximum norm on continuous function space $C(\mathcal{D})$. For $A = (a_{ij})$, a matrix of dimension $d \times r$, denote by $\|A\|_{\mathrm{F}}^2 = \sum_{i=1}^d \sum_{j=1}^r a_{ij}^2$ the Frobenius norm. Denote $\max(a, 0)$ by $(a)^+$. Denote $\boldsymbol{x} \cdot \boldsymbol{y}$ as the inner product of two vectors.

### A.2.1 PROOF OF THEOREM 3.2

**Theorem A.1 (Equivalence between Guided Flow Matching and Lyapunov Control)** *For the VF $\boldsymbol{u}_t(\boldsymbol{x}_t)$ that generates the probability path $p_t(\boldsymbol{x})$, finding the guidance $c_t(\boldsymbol{x}_t)$ to the vector field $\boldsymbol{u}_t(\boldsymbol{x}_t)$ to perform conditional sampling $p_t'(\boldsymbol{x}) = \frac{1}{Z_t} p_t(\boldsymbol{x}) \mathrm{e}^{-J(\boldsymbol{x})}$ is equivalent to finding the controller that satisfies the local Lyapunov condition, where energy function $J(\boldsymbol{x})$ contributes to Lyapunov function as $V \propto -J$, e.g., $V = -J$.*

**Proof.** Let $p_t$ be the probability path governed by the continuity equation

$$\partial_t p_t(\boldsymbol{x}) + \nabla \cdot \big( p_t(\boldsymbol{x})\, \boldsymbol{u}_t(\boldsymbol{x}) \big) = 0, \tag{4}$$

where $\boldsymbol{u}_t$ is the trained flow model. For energy $J(\boldsymbol{x})$, define the reweighted path as

$$p_t'(\boldsymbol{x}) = \frac{1}{Z_t}\, e^{-J(\boldsymbol{x})}\, p_t(\boldsymbol{x}), \qquad Z_t = \int e^{-J(\boldsymbol{x})}\, p_t(\boldsymbol{x})\, \mathrm{d}\boldsymbol{x}. \tag{5}$$

We assume sufficient smoothness, and either periodic boundary conditions or vanishing flux at infinity so that boundary integrals of divergences vanish.

**Step 1.** By definition equation 5 we have,

$$\begin{aligned}
\partial_t p_t' = \partial_t\Big(\frac{e^{-J}}{Z_t} p_t\Big) &= \frac{e^{-J}}{Z_t}\, \partial_t p_t + p_t\, \partial_t\Big(\frac{e^{-J}}{Z_t}\Big) \\
&= \frac{e^{-J}}{Z_t}\, \partial_t p_t - \big(\partial_t \log Z_t\big) \underbrace{\frac{e^{-J}}{Z_t} p_t}_{=\, p_t'} \quad (J \text{ independent of } t) \\
&= \frac{e^{-J}}{Z_t}\, \partial_t p_t - \big(\partial_t \log Z_t\big)\, p_t'.
\end{aligned} \tag{6}$$

Substituting the continuity equation equation 4 into equation 6 yields

$$\partial_t p_t' = - \frac{e^{-J}}{Z_t}\, \nabla \cdot (p_t \boldsymbol{u}_t) - \big(\partial_t \log Z_t\big)\, p_t'. \tag{7}$$

On the other hand, for the transport term,

$$\begin{aligned}
\nabla \cdot (p_t' \boldsymbol{u}_t) = \nabla \cdot \Big(\frac{e^{-J}}{Z_t} p_t\, \boldsymbol{u}_t\Big) &= \frac{e^{-J}}{Z_t}\, \nabla \cdot (p_t \boldsymbol{u}_t) + (p_t \boldsymbol{u}_t) \cdot \nabla\Big(\frac{e^{-J}}{Z_t}\Big) \\
&= \frac{e^{-J}}{Z_t}\, \nabla \cdot (p_t \boldsymbol{u}_t) - \frac{e^{-J}}{Z_t}\, p_t \boldsymbol{u}_t \cdot \nabla J = \frac{e^{-J}}{Z_t}\, \nabla \cdot (p_t \boldsymbol{u}_t) - p_t' \boldsymbol{u}_t \cdot \nabla J.
\end{aligned} \tag{8}$$

Adding equation 7 and equation 8 gives

$$\partial_t p_t' + \nabla \cdot \big(p_t' \boldsymbol{u}_t\big) = - p_t' \boldsymbol{u}_t \cdot \nabla J \; - \; \big(\partial_t \log Z_t\big)\, p_t'. \tag{9}$$

If we want $p_t'$ to be generated by the controlled field $\boldsymbol{u}_t + \boldsymbol{c}_t$, i.e.

$$\partial_t p_t' + \nabla \cdot \big(p_t'(\boldsymbol{u}_t + \boldsymbol{c}_t)\big) = 0,$$

then comparing with equation 9 we obtain the *weighted divergence equation* for $\boldsymbol{c}_t$:

$$\boxed{\nabla \cdot \big(p_t'\, \boldsymbol{c}_t\big) = p_t'\Big(\boldsymbol{u}_t \cdot \nabla J + \partial_t \log Z_t\Big).} \tag{10}$$

Integrating equation 10 over $\mathbb{R}^d$ and using divergence theorem,

$$\int_{\mathbb{R}^d} \nabla \cdot \big(p_t' \boldsymbol{c}_t\big)\, \mathrm{d}\boldsymbol{x} = 0 \quad \Longleftrightarrow \quad \int_{\mathbb{R}^d} p_t'\Big(\boldsymbol{u}_t \cdot \nabla J + \partial_t \log Z_t\Big) \mathrm{d}\boldsymbol{x} = 0.$$

The identity holds because

$$\begin{aligned}
\partial_t \log Z_t &= \frac{1}{Z_t} \partial_t \int e^{-J} p_t\, \mathrm{d}\boldsymbol{x} = \frac{1}{Z_t} \int e^{-J} \partial_t p_t\, \mathrm{d}\boldsymbol{x} = -\frac{1}{Z_t} \int e^{-J} \nabla \cdot (p_t \boldsymbol{u}_t)\, \mathrm{d}\boldsymbol{x} \\
&= -\frac{1}{Z_t} \int \nabla \cdot \big(e^{-J} p_t \boldsymbol{u}_t\big)\, \mathrm{d}\boldsymbol{x} + \frac{1}{Z_t} \int (p_t \boldsymbol{u}_t) \cdot \nabla(e^{-J})\, \mathrm{d}\boldsymbol{x} \\
&= \frac{1}{Z_t} \int (p_t \boldsymbol{u}_t) \cdot \big(-e^{-J} \nabla J\big)\, \mathrm{d}\boldsymbol{x} = -\int p_t' \boldsymbol{u}_t \cdot \nabla J\, \mathrm{d}\boldsymbol{x},
\end{aligned}$$

where the divergence integral vanishes by the boundary assumption. Hence the right-hand side of equation 10 has zero integral and equation 10 admits solutions $\boldsymbol{c}_t$ (e.g. $\boldsymbol{c}_t = \nabla \psi_t$ with a weighted Neumann problem for $\psi_t$).

**Step 2: Constructing a Lyapunov-compatible control.** We claim that a solution $c_t$ to equation 10 can be chosen so that $J$ relates to a local Lyapunov function as $V = -J$ under the controlled flow $u_t + c_t$. Decompose

$$c_t(x) = c_t^\perp(x) + c_t^\top(x), \qquad c_t^\perp(x) = \alpha_t(x)\frac{\nabla V(x)}{\|\nabla V(x)\|}, \quad \nabla V(x)\cdot c_t^\top(x) \equiv 0,$$

with the convention $\alpha_t(x) = 0$ at critical points $\{x : \nabla V(x) = 0\}$. Choose

$$\alpha_t(x) = -\frac{u_t(x)\cdot \nabla V(x)}{\|\nabla V(x)\|} - \delta_t(x)\frac{V}{\|\nabla V(x)\|}, \qquad \delta_t(x) \geq 0.$$

Then along any trajectory $x_t$ driven by $u_t + c_t$,

$$\frac{\mathrm{d}}{\mathrm{d}t}V(x_t) = \nabla V(x_t)\cdot\big(u_t(x_t) + c_t(x_t)\big) = -\gamma_t(x_t)V(x_t),$$

so $V$ is locally Lyapunov. The tangential component $c_t^\top$ is then chosen to satisfy the residual of equation 10:

$$\nabla\cdot\big(p_t' c_t^\top\big) = p_t'\big(u_t\cdot\nabla J + \partial_t \log Z_t\big) - \nabla\cdot\big(p_t' c_t^\perp\big),$$

which admits solutions $c^\top$ because both sides have zero integral.

**Step 3: From Lyapunov control to guided control.** Conversely, suppose there exists a control $c_t$ with

$$V = J, \nabla V(x)\cdot\big(u_t(x) + c_t(x)\big) \leq -\delta V.$$

Define

$$\phi_t(x) := \partial_t \log Z_t + u_t(x)\cdot \nabla V(x),$$

and the guided control $\tilde{c}$ should be a solution of

$$\nabla\cdot\big(p_t' \tilde{c}_t\big) = p_t'\,\phi_t.$$

To enforce the Lyapunov control can also satisfy the divergence equation, we correct the Lyapunov control as $\tilde{c} = c + w$, where $w$ obeys to the following equations:

$$\nabla V(x)\cdot w_t(x) = 0, \ \forall x \in O(x^*, \varepsilon), \text{ for each local minima } x^*, \tag{11}$$

$$\nabla\cdot(p_t' w_t) = p_t'\phi_t - \nabla\cdot(p_t' c_t). \tag{12}$$

By Step 1, equation equation 12 is solvable under the mild boundary condition $\int_{\mathbb{R}^d} \nabla\cdot(p_t' w_t)\,\mathrm{d}x = 0$. The purpose of equation equation 11 is to ensure that the modified control $\tilde{c}_t$ satisfies the Lyapunov decrease condition. Since equation 11 is only required to hold in a small neighborhood of the local minimum of $V$, there remains substantial freedom to adjust $w_t$ by adding any element of the nullspace of the weighted divergence operator $L_t(\cdot) = \nabla\cdot(p_t'\cdot)$.

Steps 2 and 3 provide two opposite directions for constructing a control (or guidance term) that simultaneously satisfies both the Lyapunov inequality and the divergence constraint equation 10. However, the proof reveals that starting from a Lyapunov-stable control and attempting to enforce the divergence equation requires solving the coupled PDEs equation 11–equation 12, which is generally more difficult. In practice, it is considerably easier to begin with a guidance term that already satisfies the divergence equation equation 10, and then impose Lyapunov stability via the projection operation in Theorem 4.1.

The above derivations can easily be applied to the case $V = kJ$ with $k > 0$.

Combing the above three steps, we complete the proof. ∎

### A.2.2 Proof and Analysis of Proposition 3.3

**Proposition A.2** *The following commonly used guidance strategies in generative modeling can all be interpreted as Lyapunov control within our unified framework:*

- ***Classifier Guidance***: *Given a trained classifier $p(y|x)$, the Lyapunov function for guided distribution $p(x|y)$ specified on conditioner $y$ is $V_y(x) = \log p(y|x)$.*

- **Reward Guidance**: *In reinforcement learning tasks with reward function $R(\boldsymbol{x})$, the Lyapunov function for guided distribution $\frac{1}{Z} p_t(\boldsymbol{x}) e^{R(\boldsymbol{x})}$ concentrates probability mass in high-reward regions is $V(\boldsymbol{x}) = -R(\boldsymbol{x})$.*

- **Energy-Based Model (EBM) Guidance**: *For a target EBM $p(\boldsymbol{x}) \propto e^{-E(\boldsymbol{x})}$, the Lyapunov function is naturally $V(\boldsymbol{x}) = -E(\boldsymbol{x})$.*

- **Image inverse problems.** *Let the forward operator be $\boldsymbol{y} = H(\boldsymbol{x}) + \varepsilon$ with $\varepsilon \sim \mathcal{N}(0, \sigma^2 I)$. Then $p(\boldsymbol{y}|\boldsymbol{x}) \propto \exp\left(-\frac{1}{2\sigma^2} \|H(\boldsymbol{x}) - \boldsymbol{y}\|_2^2\right)$ and a natural Lyapunov function is $V_{\boldsymbol{y}}(\boldsymbol{x}) = \frac{1}{2\sigma^2} \|H(\boldsymbol{x}) - \boldsymbol{y}\|_2^2$, yielding guided sampling $p'_t(\boldsymbol{x}) \propto p_t(\boldsymbol{x}) \exp\left(-V_{\boldsymbol{y}}(\boldsymbol{x})\right)$ that enforces data consistency Song et al. (2023).*

**Proof.**

- **Classifier Guidance and image inverse problems.**: Taking $J(x) = -\log p(y|x)$, we compute

$$\frac{1}{Z} p_t(x) e^{-J(x)} = \frac{p_t(x) p(y|x)}{\int p_t(x) p(y|x)\, dx} = \frac{p_t(x) \frac{p(x,y)}{p(x)}}{\int p_t(x) \frac{p(x,y)}{p(x)}\, dx} = p(x|y).$$

Thus the guided distribution is exactly conditional sampling.

- **Reward Guidance**: With $J(x) = -R(x)$, the guided law is

$$p'_t(x) = \frac{1}{Z} p_t(x) e^{R(x)},$$

which emphasizes regions of higher reward. This is equivalent to Lyapunov control with $V(x) = -R(x)$, where the control term reduces $V(x)$ along the trajectory, steering toward reward-maximizing states.

- **EBM Guidance**: If the target distribution is $p(x) \propto e^{-E(x)}$, then

$$p'_t(x) = \frac{1}{Z} p_t(x) e^{-E(x)},$$

which matches the EBM density. Interpreting $V(x) = E(x)$ as Lyapunov function, the control enforces descent in $V(x)$, aligning the flow with low-energy modes.

∎

### A.2.3    PROOF OF THEOREM 4.1

**Theorem A.3 (Lyapunov Guarantee for Guidance)** *For a candidate controller $\boldsymbol{c}$ and the guidance controller space $\mathcal{U}(V) = \{\boldsymbol{u} : \nabla V \cdot (\boldsymbol{u}_t + \boldsymbol{c}_t) + \delta V \leq 0\}$ that rigorously satifies the local Lyapunov condition in Proposition 3.1, define the projection operator as*

$$\pi(\boldsymbol{c}_t, \mathcal{U}(V)) \triangleq \boldsymbol{c}_t - \frac{\max\left(0, \nabla V(\boldsymbol{x}) \cdot (\boldsymbol{u}_t(\boldsymbol{x}) + \boldsymbol{c}_t(\boldsymbol{x})) + \delta V(\boldsymbol{x})\right)}{\|\nabla V(\boldsymbol{x})\|^2} \nabla V(\boldsymbol{x}).$$

*Then $\pi(\boldsymbol{c}_t, \mathcal{U}(V))$ is locally Lipschitz continuous and thus the guided flow under $\pi(\boldsymbol{c}_t, \mathcal{U}(V))$ is well defined, and $\pi(\boldsymbol{c}_t, \mathcal{U}(V)) \in \mathcal{U}(V)$.*

**Proof.** The local Lipschitz continuity of the projected function naturally comes from the local Lipschitz continuity of the considered functions $\boldsymbol{u}$, $\boldsymbol{c}$ and $V$. We directly check the inequality constraint in $\mathcal{U}(V)$ is satisfied by the projection element, that is

$$\mathcal{L}_{\boldsymbol{u}+\boldsymbol{c}^*} V\big|_{\boldsymbol{c}^* = \pi(\boldsymbol{c}, \mathcal{U}(V))} \leq \delta - V.$$

Since the controller has affine actuator, from the definition of the Lie derivative operator, we have

$$\mathcal{L}_{\boldsymbol{u}+\boldsymbol{c}^*} V\big|_{\boldsymbol{c}^* = \pi(\boldsymbol{c}, \mathcal{U}(V))} = \nabla V \cdot \left(\boldsymbol{u} + \boldsymbol{c} - \frac{\max(0, \mathcal{L}_{\boldsymbol{u}+\boldsymbol{c}} V + \delta V)}{\|\nabla V\|^2} \cdot \nabla V\right)$$

$$= \nabla V \cdot (\boldsymbol{u} + \boldsymbol{c}) - \nabla V \cdot \frac{\max(0, \mathcal{L}_{\boldsymbol{u}+\boldsymbol{c}} V + \delta V)}{\|\nabla V\|^2} \cdot \nabla V$$

$$= \mathcal{L}_{\boldsymbol{u}+\boldsymbol{c}} V - \max(0, \mathcal{L}_{\boldsymbol{u}+\boldsymbol{c}} V + \delta V) \leq -\delta V.$$

∎

### A.2.4 Global stability vs local stability

We would like to clarify the notion of "global optimization" in this context. In control theory, the relevant distinction is between global stability (a single global equilibrium acting as an attractor) and local stability (multiple equilibria, each with its own local basin). Our work follows the latter, because the target distribution in generative modeling naturally contains multiple modes, each corresponding to a different local minimizer of the potential $V$. In such multi-modal settings, global stability is neither realistic nor desirable: enforcing a single global attractor would collapse all modes. Hence Lyapunov-based methods must (and should) focus on local stability around each mode, ensuring that trajectories contract when they enter the basin of the corresponding data mode. This is exactly what LyaGuide leverages. The Lyapunov function is used to guarantee local contraction toward the correct mode, not to solve a global optimization problem. Of course, if the target distribution contains only a single minimizer, then our pseudo-projection mechanism automatically enforces global stability. We have therefore added a Theorem in the revised manuscript to explicitly state and prove this global guarantee in the single-equilibrium case. Therefore, the use of a local Lyapunov function is not a limitation of our method but a direct reflection of the structure of the underlying generative problem. Moreover, as shown in Fig. 5, the resulting local contraction significantly accelerates convergence and reduces W-2 distance across all inference times.

### A.2.5 Intuitive explanation of the theory

The Theorem 3.2 reveals a useful conceptual symmetry between guidance-compatible controls (those satisfying the weighted divergence equation) and Lyapunov-stable controls (those ensuring monotone contraction of the Lyapunov function). In principle, starting from a Lyapunov-stable control, one can always construct a guidance-compatible control by solving an appropriate correction equation so that the divergence constraint equation 10 is satisfied. Conversely, given any guidance term, one can adjust its normal component along $\nabla V$ to enforce Lyapunov decrease while leaving the divergence structure intact.

However, these two directions are not equally tractable. The conversion from Lyapunov stability to guidance requires solving coupled PDE system equation 11,equation 12 that simultaneously enforces tangency to the level sets of $V$ and corrects the weighted divergence. This construction is mathematically valid but typically costly and impractical for high-dimensional learning problems.

The reverse direction is substantially simpler and forms the basis of LyaGuide. Modern flow-matching or score-based models already produce guidance terms that satisfy the divergence equation by construction. Given such a guidance field, Theorem 4.1 shows that we can impose Lyapunov stability without solving any PDE by projecting the field onto the Lyapunov-stable set at each point. The projection adjusts only the normal component of the guidance vector along $\nabla V$, guaranteeing strict decay of sampling trajectory along $V$ while preserving the original divergence structure.

This two-way relationship explains why LyaGuide is both principled and efficient: the theory ensures that stable and divergence-compatible controls are mutually reachable, whereas the algorithm exploits the easy direction—projecting guidance to stability—to obtain a control that satisfies both properties simultaneously.

### A.2.6 Further discussions

**Both stable and generating the guided probability path.** The results in Theorem 3.2 characterize the existence of a control/guidance that simultaneously satisfies: (i) the weighted continuity equation (Eq. equation 10 in Appendix) that generates the guided probability path, (we denote such controllers form a space $\mathcal{U}_g$) and (ii) the local Lyapunov decrease condition (such controllers form a space $\mathcal{U}_s$). The pseudo-projection in Theorem 4.1 is designed exclusively to enforce the Lyapunov inequality. Since the projection is a pointwise modification of the vector field, it does not impose any constraint on the weighted divergence term $\nabla \cdot (p'_t u_t)$, and thus does not preserve the weighted continuity equation in general. Our idea is to project a guidance term $c \in \mathcal{U}_g$ into the subspace $\mathcal{U}_g \cap \mathcal{U}_s$ to enforce stability for the guidance term, which thereby accelerating the sampling process and enhancing the sampling robustness. How to develop a projection operator or an alternative correction that enforces *both* Lyapunov stability and the weighted divergence constraint remains an open problem and a promising direction for future work.

### A.3 RELATED WORK

**Meta-learning and Task-specific Adaptation Modules.** Meta-learning aims to enable models to generalize rapidly to new tasks with limited data or fine-tuning. Among various approaches, a subset of methods avoids modifying the entire network and instead introduces lightweight task-adaptive modules that guide inference without retraining the base model. Representative examples include feature-wise transformations such as FiLM layers, task-conditioning adapters, and modular meta-learning architectures (Finn et al., 2017; Rusu et al., 2019; Rebuffi et al., 2017). These methods typically operate within a meta-training/meta-testing paradigm over a distribution of tasks. In contrast, our method focuses on post-training adaptation of generative flows for new conditioning objectives, by introducing Lyapunov-inspired correction terms without retraining or task-specific fine-tuning. While sharing the spirit of lightweight adaptation, our approach is grounded in control-theoretic stability principles and ensures rigorous guarantees through a projection scheme, offering a theoretically justified alternative to heuristic task adapters.

**Guidance in Generative Models.** In diffusion models, classifier guidance (Dhariwal & Nichol, 2021) and classifier-free guidance (Ho & Salimans) are two influential approaches; further extensions include guidance from reward/EBM priors (Janner et al., 2022; Zhang & Xu, 2023; LeCun et al., 2006). For flow matching, recent work studies guidance at the field level and shows how importance reweighting on the joint induces pathwise changes to the marginals (Feng et al., 2025). In parallel, Sprague et al. (Sprague et al., 2024) proposed incorporating stochastic stability into flow matching, which is highly relevant to our Lyapunov control formulation. Compared with their stability-oriented modifications, our framework provides a unified *control-theoretic* view that connects diverse guidance strategies and introduces a pseudo projection operator to rigorously enforce Lyapunov inequalities at inference. Furthermore, Albergo et al. (Albergo et al., 2023) and Chen et al. (Chen et al., 2023) proposed alternative generative formulations via stochastic interpolants and phase-space bridges, respectively. While sharing a control-inspired spirit, these approaches differ fundamentally from ours: they reformulate the generative process itself, whereas we provide a general-purpose guidance scheme applicable to pre-trained flows.

**Learning Certificates.** Score-based modeling interprets $\nabla \log p(\boldsymbol{x}) = -\nabla V(\boldsymbol{x})$ as a learnable vector field (Song & Ermon, 2019; Song et al., 2021), while EBMs directly parametrize an energy (LeCun et al., 2006; Du & Mordatch, 2019). Few-shot or preference-based specification of priors has been explored for RL and diffusion guidance (Janner et al., 2022; Lee et al., 2025), but typically without Lyapunov certificates. Liu et al. (Liu et al., 2023a) studied physics-informed neural networks for learning and verifying Lyapunov functions in PDE systems, and Kang et al. (Kang et al., 2021) developed stable neural ODEs with Lyapunov-stable equilibria. Both are conceptually close to our Scenario 2, where we *learn* a Lyapunov candidate $V_\theta$ from sparse supervision and *jointly* enforce Lyapunov inequalities during training. Our method complements these works by focusing specifically on generative modeling and introducing an efficient projection-based stabilization scheme.

**Generative Modeling in Latent or Alternative Spaces.** Other approaches focus on modifying the generative space. Dockhorn et al. (Dockhorn et al., 2022) proposed score-based generative modeling in latent spaces, which reduces computational cost and enables better representation learning. Such latent guidance methods are complementary to our work: while they change the domain of generative dynamics, our framework provides a universal stabilizing principle applicable regardless of the space in which flows are defined.

### A.4 NOTES ON ROBUSTNESS OF THE INITIAL VALUE

In this section, we discuss the influence of the initial value to the guided generative modelling. We note that in the original flow model, the initial data is sampled from the initial distribution $p_0$. For the guided flow, we still sample the initial value from $p_0$, which is common in flow matching community Lipman et al. (2023); Tong et al. (2023); Feng et al. (2025). However, according to the continuity equation, to generate $p'_t$ in Theorem 3.2, the initial distribution should be $p'_0 = 1/Z_0 e^{-J(\boldsymbol{x})} p_0$ instead of $p_0$. So we investigate how the choice of initial distribution influence the generation effect from the perspective of distribution convergence.

**Theorem A.4 (Robustness to Initial Distribution)** *Let $u_t(x)$ be the learned vector field and $c_t(x)$ the guidance control derived from Lyapunov function $V(x)$. Assume the guided field $f_t(x) = u_t(x) + c_t(x)$ satisfies the local Lyapunov condition with rate $\delta > 0$. Then, for any two initial distributions $p_0$ and $p_0^\star$, the corresponding marginals $p_t$ and $p_t^\star$ along the flow satisfy*

$$W_2(p_t, p_t^\star) \leq e^{-\delta t}\, W_2(p_0, p_0^\star), \quad \forall t \in [0, 1].$$

*In particular, even if the actual inference starts from a different initial law $p_0$ (e.g. Gaussian noise) instead of the theoretical $p_0^\star$, the terminal distribution at $t = 1$ remains exponentially close to the desired guided distribution $p_1^\star \propto q(x)e^{-J(x)}$.*

**Lemma A.5 (Lyapunov-guided field implies contraction)** *Let $f_t(x) = u_t(x) + c_t(x)$ with a Lyapunov-based guidance $c_t(x) = -K_t(x)\nabla V(x)$, where $V \in \mathcal{C}^2$ and $K_t(x) \in \mathbb{R}^{d \times d}$ is symmetric positive definite. Fix a forward-invariant sublevel set $\mathcal{S}_\rho := \{x : V(x) \leq \rho\}$. Assume on $\mathcal{S}_\rho$:*

1. ***Strong convexity of*** $V$***:*** $\nabla^2 V(x) \succeq mI$ *for some $m > 0$;*

2. ***Uniform gain:*** $K_t(x) \succeq \kappa I$ *for some $\kappa > 0$;*

3. ***Bounded symmetric Jacobian of*** $u_t$***:*** $\frac{1}{2}\bigl(\nabla u_t(x) + \nabla u_t(x)^\top\bigr) \preceq LI$ *for some $L \in \mathbb{R}$;*

4. ***(Optional) Slowly-varying gain:*** $\|\nabla K_t(x)\| \leq B$ *(if $K_t$ depends on $x$).*

*Then, for all $x, y \in \mathcal{S}_\rho$ and $t \in [0, 1]$,*

$$\langle x - y,\ f_t(x) - f_t(y)\rangle \ \leq\ -\delta\,\|x - y\|^2,$$

*with*

$$\delta \ =\ \kappa m - L - \varepsilon, \qquad \varepsilon := \begin{cases} 0, & \text{if } K_t \text{ is constant in } x; \\ B\,\sup_{z \in \mathcal{S}_\rho} \|\nabla V(z)\|, & \text{otherwise.} \end{cases}$$

*In particular, if $\kappa m > L + \varepsilon$, the contraction condition equation 13 holds on $\mathcal{S}_\rho$.*

**Proof.** By the mean-value integral along the segment $\gamma(\theta) = y + \theta(x - y)$,

$$f_t(x) - f_t(y) = \left(\int_0^1 \nabla f_t(\gamma(\theta))\, d\theta\right)(x - y).$$

Hence

$$\langle x - y,\ f_t(x) - f_t(y)\rangle = \int_0^1 \langle x - y,\ \operatorname{sym} \nabla f_t(\gamma(\theta))\,(x - y)\rangle d\theta,$$

where $\operatorname{sym} A = \frac{1}{2}(A + A^\top)$. It suffices to upper bound $\operatorname{sym} \nabla f_t$. Compute

$$\nabla f_t = \nabla u_t - \nabla\bigl(K_t \nabla V\bigr) = \nabla u_t - \bigl[(\nabla K_t)\nabla V^\top + K_t \nabla^2 V\bigr].$$

Taking symmetric parts yields

$$\operatorname{sym} \nabla f_t \ \preceq\ LI \ -\ K_t \nabla^2 V \ +\ \operatorname{sym}\bigl((\nabla K_t)\nabla V^\top\bigr).$$

On $\mathcal{S}_\rho$, by (A1)–(A3): $K_t \nabla^2 V \succeq \kappa m I$ and $\operatorname{sym}((\nabla K_t)\nabla V^\top)$ has operator norm $\leq \|\nabla K_t\|\,\|\nabla V\| \leq B\,\sup_{\mathcal{S}_\rho} \|\nabla V\|$. Therefore

$$\operatorname{sym} \nabla f_t \ \preceq\ \bigl(L - \kappa m + \varepsilon\bigr)I.$$

Integrating along $\gamma(\theta)$ gives $\langle x - y, f_t(x) - f_t(y)\rangle \leq -(\kappa m - L - \varepsilon)\|x - y\|^2$, which proves the claim with $\delta = \kappa m - L - \varepsilon$. ∎

**Theorem A.6 (Robustness to the Initial Distribution)** *Assume $c_t = -K_t \nabla V$ and conditions (A1)–(A4) in Lemma A.5 hold on a forward-invariant sublevel set; then equation 13 follows with $\delta = \kappa m - L - \varepsilon$.*

*Let $u_t$ be the learned vector field and $c_t$ the guidance control derived from a Lyapunov function $V$, and denote the guided field by $f_t := u_t + c_t$. Assume there exists $\delta > 0$ and a domain $\mathcal{D} \subset \mathbb{R}^d$ that is forward invariant under $f_t$ such that*

$$\langle x - y,\ f_t(x) - f_t(y) \rangle \ \leq\ -\delta \, \|x - y\|^2, \qquad \forall\, x, y \in \mathcal{D},\ \forall t \in [0, 1]. \tag{13}$$

*Let $p_t$ and $p_t^\star$ be the marginals obtained by pushing forward $p_0$ and $p_0^\star$ along the flow of $f_t$, respectively. Then, for all $t \in [0, 1]$,*

$$W_2(p_t, p_t^\star) \ \leq\ e^{-\delta t} \, W_2(p_0, p_0^\star).$$

*In particular, taking $p_0^\star$ as the "theoretical" initialization (e.g. the marginal induced by the conditional path), any inference initialized from a different prior $p_0$ (e.g. a Gaussian) remains exponentially close to the target guided law at $t = 1$.*

**Proof.** Let $\Phi_{t,0} : \mathcal{D} \to \mathcal{D}$ denote the flow map associated with the ODE $\dot{x} = f_t(x)$, i.e., $x_t = \Phi_{t,0}(x_0)$. Fix any two initial states $x_0, y_0 \in \mathcal{D}$ and consider the distance $D(t) := \frac{1}{2}\|x_t - y_t\|^2$ where $x_t = \Phi_{t,0}(x_0)$ and $y_t = \Phi_{t,0}(y_0)$. Then

$$\dot{D}(t) = \langle x_t - y_t,\ f_t(x_t) - f_t(y_t) \rangle \ \leq\ -\delta \, \|x_t - y_t\|^2 = -2\delta \, D(t),$$

where we used equation 13. By Grönwall's inequality,

$$\|x_t - y_t\| \ \leq\ e^{-\delta t} \, \|x_0 - y_0\|, \qquad \forall t \in [0, 1].$$

Now let $\gamma_0$ be any coupling in $\Pi(p_0, p_0^\star)$. Push it forward through the product flow to obtain a coupling $\gamma_t := (\Phi_{t,0} \times \Phi_{t,0})_\# \gamma_0 \in \Pi(p_t, p_t^\star)$. Then

$$\int \|x - y\|^2 \, d\gamma_t(x, y) = \int \left\| \Phi_{t,0}(x_0) - \Phi_{t,0}(y_0) \right\|^2 d\gamma_0(x_0, y_0) \ \leq\ e^{-2\delta t} \int \|x_0 - y_0\|^2 \, d\gamma_0(x_0, y_0).$$

Taking the infimum over all initial couplings $\gamma_0 \in \Pi(p_0, p_0^\star)$ yields

$$W_2^2(p_t, p_t^\star) \ \leq\ e^{-2\delta t} \, W_2^2(p_0, p_0^\star),$$

and the stated bound follows by taking square roots. ∎

**Remark A.7** (**Link to Lyapunov condition**) *The contractivity assumption equation 13 is implied locally if the Lyapunov control is taken as $c_t(x) = -K_t(x)\nabla V(x)$ with $K_t(x) \succeq \kappa I$, and $V$ is $m$-strongly convex on a forward-invariant sublevel set (so that $\nabla^2 V \succeq m I$ there), while the symmetric part of $\nabla u_t$ is bounded above by $L_t I$ with $\kappa m - L_t \geq \delta > 0$. Then $\frac{1}{2}\left(\nabla f_t + \nabla f_t^\top\right) \preceq -\delta I$, which is equivalent to equation 13.*

This result formalizes the robustness of guided flow matching with respect to the choice of the initial distribution. Although the theoretical formulation involves the marginal initialization $p_0^\star = \int p_0(x \mid x_1) q(x_1)\, dx_1$, in practice one typically samples directly from a Gaussian prior $p_0$. The contraction guaranteed by the Lyapunov condition ensures that the discrepancy between $p_0$ and $p_0^\star$ vanishes exponentially fast along the trajectory. Consequently, by the terminal time $t = 1$, the generated distribution $p_1$ is already arbitrarily close to the target law, making inference from $p_0$ both valid and effective.

## A.5 EXPERIMENTAL CONFIGURATIONS AND ADDITIONAL EXPERIMENTS

### A.5.1 SYNTHETIC DATA EXPERIMENTS.

For the 2D synthetic benchmarks (uniform-to-8Gaussians, circle-to-S-curve and 8Gaussians-to-Moons), we follow the setup of (Feng et al., 2025). The flow model is trained with displacement interpolation and a 4-layer MLP with hidden dimension 128, using Adam optimizer with learning rate $10^{-4}$. Each experiment is trained for 20k iterations with batch size 512. Guidance baselines include Monte Carlo ($g^{\text{MC}}$), covariance-based guidance ($g^{\text{cov-A}}$, $g^{\text{cov-G}}$), contrastive energy guidance (CEG), and learned guidance $g_\phi$. For each method, we apply our pseudo projection operation (LyaGuide-ES/AS/CS) in the inference stage.

**Few-shot Supervision (Scenario 2).** In Scenario 2, we subsample preference data–score pairs with sizes varying from 128 to 1024. The Lyapunov candidate $V_\theta$ is parameterized as an MLP with 3

hidden layers (width 64) and trained using weighted regression. We emphasize low-score samples to encourage convexity around minima. Once $V_\theta$ is learned, the corresponding guidance control is synthesized using either explicit descent or integrated into $g_\phi$.

**Ablation study.** In this section we investigate the effects of the parameters $\delta, k$ to the performance of LyaGuide. The parameter $\delta$ appears both in the Lyapunov convergence rate in Proposition 3.1 and in the explicit projection term in Theorem 4.1. This dual influence makes $\delta$ a global hyperparameter affecting all initial methods when incorporated into LyaGuide. From Figure A.5.1, we observe that the method is overall robust to $\delta$ across a broad range of values. As $\delta$ increases, the generated samples concentrate more strongly near the minima of the Lyapunov function $V$, i.e., the low-energy region. This stronger contraction improves convergence but also reduces the probability of visiting low-density regions, thereby decreasing exploration capability. In other words, a large $\delta$ induces over-contraction, which may excessively bias sampling towards high-density areas. Therefore, $\delta$ can be tuned according to practical needs: a smaller $\delta$ enhances exploration, while a larger $\delta$ yields stronger stability and contraction. However, as shown in both Figure A.5.1 and Table 3, excessively large $\delta$ may cause LyaGuide to underperform compared to the original guidance. Based on our empirical findings, a practical recommendation is to choose $\delta \in [0, 1]$, which consistently yields a good balance between exploration and stability.

**Sensitivity with respect to $k$ (gradient-based guidance only).** The control coefficient $k$ only affects gradient-based guidance methods (e.g., energy- or score-based guidance), since it appears in the initial guidance term $-k\nabla V$ prior to applying LyaGuide. Therefore, we evaluate $k$ exclusively on gradient-based variants, as reported in Table 3. The results show that increasing $k$ generally improves sample quality, as stronger control in the Lyapunov direction facilitates more effective contraction of the gradient flow. This benefit persists across different $\delta$ values, although extreme $\delta$ can diminish the improvement. These observations confirm that $k$ acts as a refinement factor for stabilising gradient-based methods, rather than as a global sensitivity parameter.

**Brief Summary.** The parameter $\delta$ influences both Lyapunov convergence and the projection step, and the method exhibits low sensitivity to its variation. Larger $\delta$ improves convergence but reduces exploration; thus, we recommend choosing $\delta \in [0, 1]$. The parameter $k$ affects only gradient-based guidance, and larger $k$ generally improves performance, as shown in Table 3. We thank the reviewer again for this valuable comment. The new ablation studies have been incorporated into the revised manuscript.

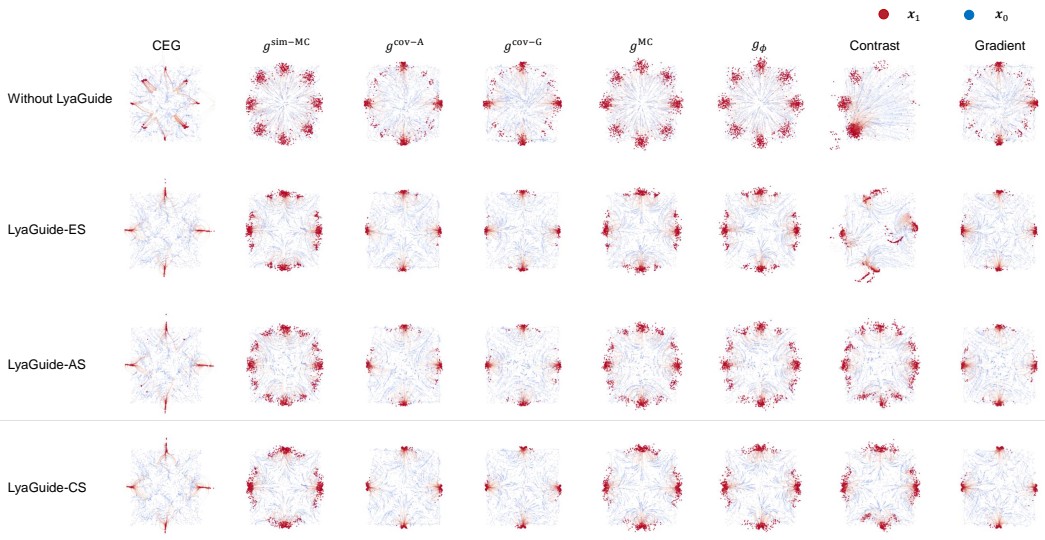

Figure 6: Scenario 1 results in 8-Gaussian task with different variants of LyaGuide.

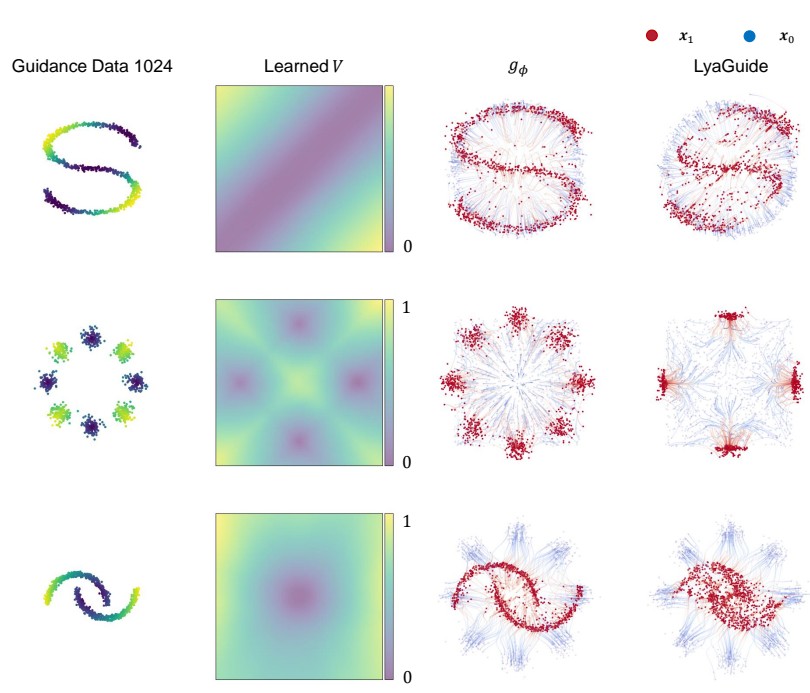

Figure 7: Scenario 2 results on synthetic data with dataset size $= 1024$.

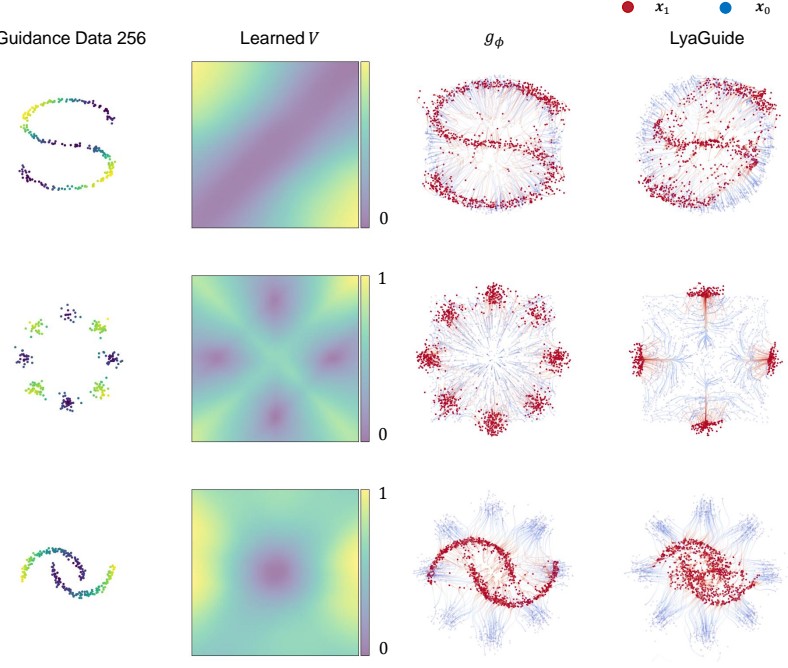

Figure 8: Scenario 2 results on synthetic data with dataset size $= 256$.

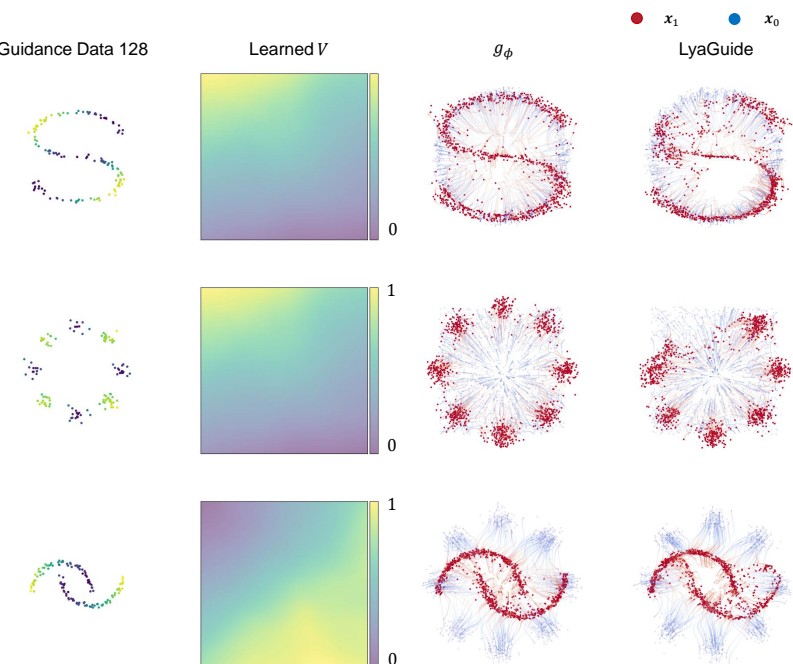

Figure 9: Scenario 2 results on synthetic data with dataset size = 128.

Table 3: Ablation study on gradient-based method in 8-Gaussian generation. Wasserstein-2 distances between the target data and the sampled data under different combinations of the hyperparameter $\delta$ and $k$ are shown.

| | Original Gradient Method | | | LyaGuide-ES | | | LyaGuide-CS | | |
|---|---|---|---|---|---|---|---|---|---|
| | $k = 0.5$ | $k = 1.0$ | $k = 1.5$ | $k = 0.5$ | $k = 1.0$ | $k = 1.5$ | $k = 0.5$ | $k = 1.0$ | $k = 1.5$ |
| $\delta = 0.2$ | 0.29 | 0.24 | 0.18 | 0.28 | 0.25 | 0.16 | 0.33 | 0.2 | 0.34 |
| $\delta = 0.4$ | 0.34 | 0.19 | 0.25 | 0.28 | 0.28 | 0.3 | 0.33 | 0.24 | **0.09** |
| $\delta = 0.6$ | 0.42 | 0.19 | 0.28 | 0.39 | 0.27 | 0.31 | 0.41 | 0.17 | 0.18 |
| $\delta = 0.8$ | 0.3 | 0.33 | 0.25 | 0.3 | 0.33 | 0.25 | 0.4 | 0.24 | 0.21 |
| $\delta = 1.0$ | 0.32 | 0.26 | 0.26 | 0.35 | 0.22 | 0.24 | 0.32 | 0.2 | 0.17 |
| $\delta = 1.5$ | 0.29 | 0.17 | 0.16 | 0.33 | 0.16 | 0.2 | 0.42 | 0.27 | 0.25 |
| $\delta = 2.0$ | 0.45 | 0.19 | 0.29 | 0.37 | 0.23 | 0.31 | 0.41 | 0.2 | 0.34 |

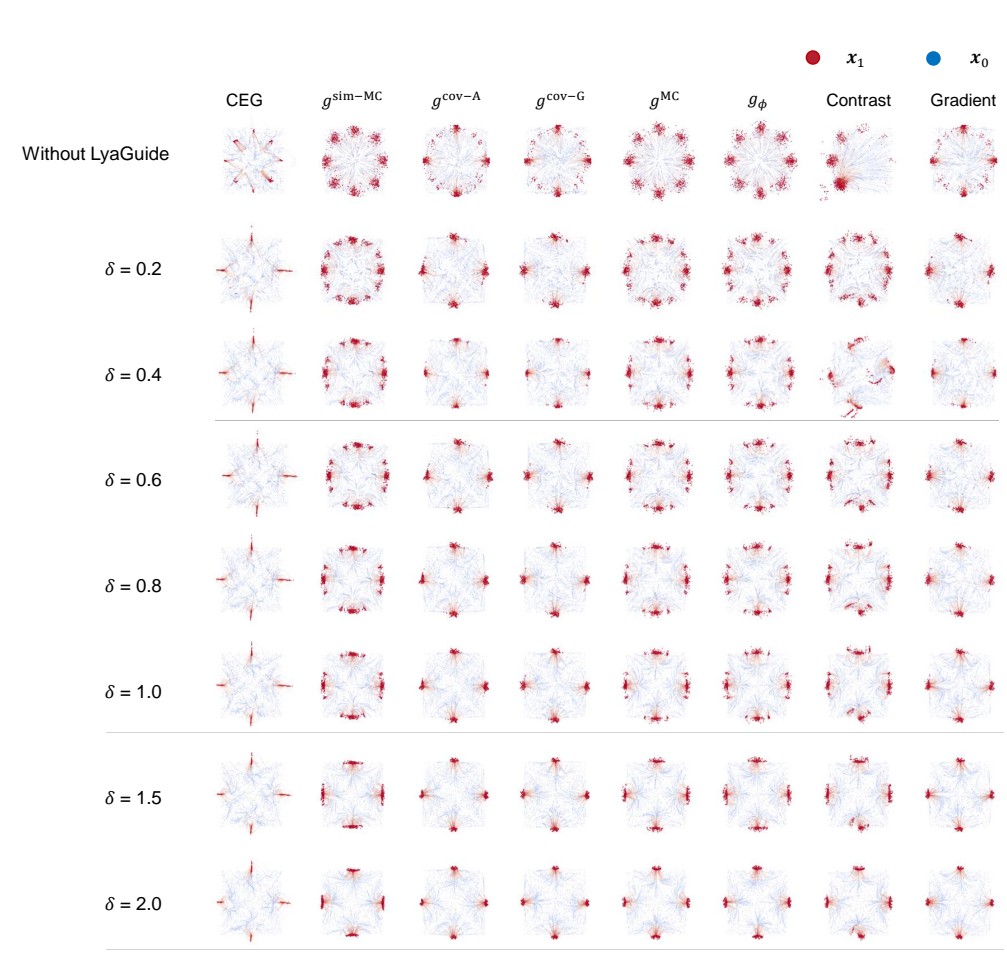

Figure 10: Ablation study in 8-Gaussian task. We investigate the effect of the Lyapunov convergence rate $\delta$ to the performance of the LyaGuide sampling. The first row corresponds to the results of the orginal methods without LyaGuide, the other rows correspond to the results of LyaGuide with different choices of $\delta$.

### A.5.2 IMAGE INVERSE PROBLEMS.

For CelebA-HQ image inpainting, deblurring, and super-resolution tasks, we adopt the same experimental protocol as (Song et al., 2023; Feng et al., 2025). Conditional flow matching (CFM) and optimal-transport CFM (OT-CFM) are used as base models. Evaluation metrics include FID, LPIPS, PSNR, and SSIM over 3000 test samples. Baselines ($g^{MC}$, $g^{cov-A}$, ΠGDM, etc.) are compared with and without LyaGuide projection. Hyperparameters for the pseudo projection are fixed across all tasks, demonstrating robustness without extra tuning.

Table 4: Image inverse problem results on CelebA-HQ (Super-Resolution task).

| | | Original Methods | | | | LyaGuide-ES | | | | LyaGuide-CS | | | |
|---|---|---|---|---|---|---|---|---|---|---|---|---|---|
| | | FID ↓ | LPIPS ↓ | PSNR ↑ | SSIM ↑ | FID ↓ | LPIPS ↓ | PSNR ↑ | SSIM ↑ | FID ↓ | LPIPS ↓ | PSNR ↑ | SSIM ↑ |
| OT-CFM | $g^{cov-A}$ | 30.2284 | 0.3713 | 22.9642 | 0.5988 | 30.2183 | 0.3712 | 22.9707 | 0.5992 | 27.5309 | 0.3630 | 22.8065 | 0.6132 |
| | $g^{sim-A}$ | 31.8224 | 0.3718 | 23.8667 | 0.6069 | 31.5397 | 0.3705 | 23.8698 | 0.6091 | 27.5309 | 0.3630 | 22.8065 | 0.6132 |
| | ΠGDM | 22.9596 | 0.2826 | 26.9492 | 0.7584 | 23.0441 | 0.2827 | 26.9462 | 0.7582 | 23.7278 | 0.2854 | 26.9099 | 0.7554 |
| | $g^{MC}$ | 24.1797 | 0.5521 | 8.7411 | 0.3628 | 24.9576 | 0.5538 | 8.8167 | 0.3637 | 25.3356 | 0.5430 | 9.0702 | 0.3848 |
| CFM | $g^{cov-A}$ | 31.9606 | 0.3769 | 22.7715 | 0.5897 | 32.0031 | 0.3769 | 22.7778 | 0.5902 | 31.2359 | 0.3741 | 22.7429 | 0.5948 |
| | $g^{sim-A}$ | 32.6209 | 0.3745 | 23.6848 | 0.6005 | 31.9223 | 0.3712 | 23.6621 | 0.6061 | 27.7175 | 0.3564 | 23.4357 | 0.6312 |
| | ΠGDM | 25.7605 | 0.2900 | 26.8810 | 0.7470 | 25.9728 | 0.2897 | 26.8811 | 0.7473 | 26.7152 | 0.2933 | 26.8322 | 0.7434 |
| | $g^{MC}$ | 26.5714 | 0.5555 | 8.9762 | 0.3588 | 28.1408 | 0.5566 | 9.0434 | 0.3595 | 27.7506 | 0.5456 | 9.3742 | 0.3836 |

Table 5: Image inverse problem results on CelebA-HQ (Gaussian deblurring task).

| | | Original Methods | | | | LyaGuide-ES | | | | LyaGuide-CS | | | |
|---|---|---|---|---|---|---|---|---|---|---|---|---|---|
| | | FID ↓ | LPIPS ↓ | PSNR ↑ | SSIM ↑ | FID ↓ | LPIPS ↓ | PSNR ↑ | SSIM ↑ | FID ↓ | LPIPS ↓ | PSNR ↑ | SSIM ↑ |
| OT-CFM | $g^{cov-A}$ | 15.7897 | 0.2738 | 26.0362 | 0.7202 | 14.9309 | 0.2699 | 26.2131 | 0.7247 | 11.9049 | 0.2534 | 26.2578 | 0.7375 |
| | $g^{sim-A}$ | 14.7355 | 0.2506 | 27.8596 | 0.7721 | 14.9201 | 0.2507 | 27.8551 | 0.7720 | 14.7920 | 0.2500 | 27.8420 | 0.7719 |
| | ΠGDM | 20.4083 | 0.2514 | 28.6943 | 0.7827 | 19.8694 | 0.2492 | 28.7049 | 0.7838 | 20.3166 | 0.2517 | 28.6720 | 0.7820 |
| | $g^{MC}$ | 24.5336 | 0.5524 | 8.7690 | 0.3640 | 28.0601 | 0.5156 | 10.5469 | 0.4110 | 27.5108 | 0.5388 | 9.2243 | 0.3913 |
| CFM | $g^{cov-A}$ | 16.6399 | 0.2830 | 25.5352 | 0.6989 | 16.0158 | 0.2792 | 25.6866 | 0.7031 | 13.1084 | 0.2641 | 25.7225 | 0.7161 |
| | $g^{sim-A}$ | 15.2263 | 0.2591 | 27.5387 | 0.7587 | 15.2263 | 0.2591 | 27.5387 | 0.7587 | 15.6122 | 0.2585 | 27.4902 | 0.7581 |
| | ΠGDM | 20.7786 | 0.2593 | 28.3883 | 0.7709 | 20.3063 | 0.2557 | 28.5493 | 0.7750 | 20.6238 | 0.2591 | 28.4349 | 0.7709 |
| | $g^{MC}$ | 26.4901 | 0.5556 | 9.0386 | 0.3612 | 31.1472 | 0.5276 | 10.3195 | 0.3959 | 30.0128 | 0.5410 | 9.5963 | 0.3917 |

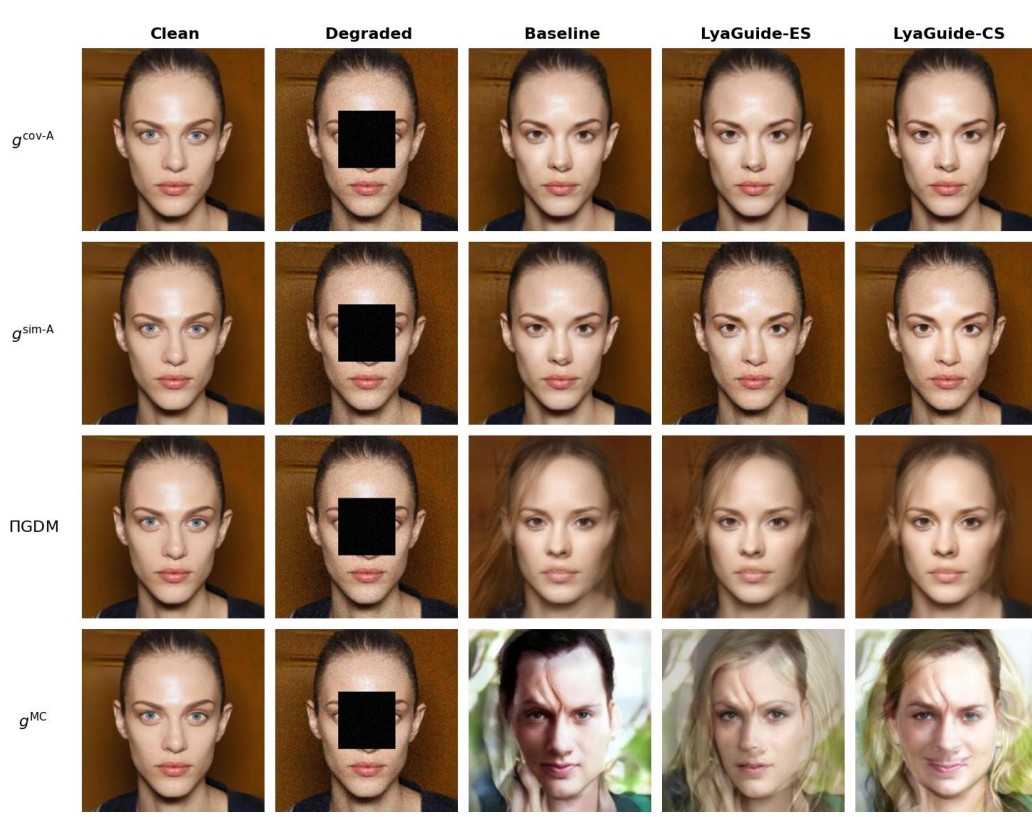

Figure 11: The visualization of the image inverse problems with the base flow matching model of mini-batch optimal transport conditional flow matching (OT-CFM). Four rows show the results of four baselines with the corresponding LyaGuide results in box-inpainting task.

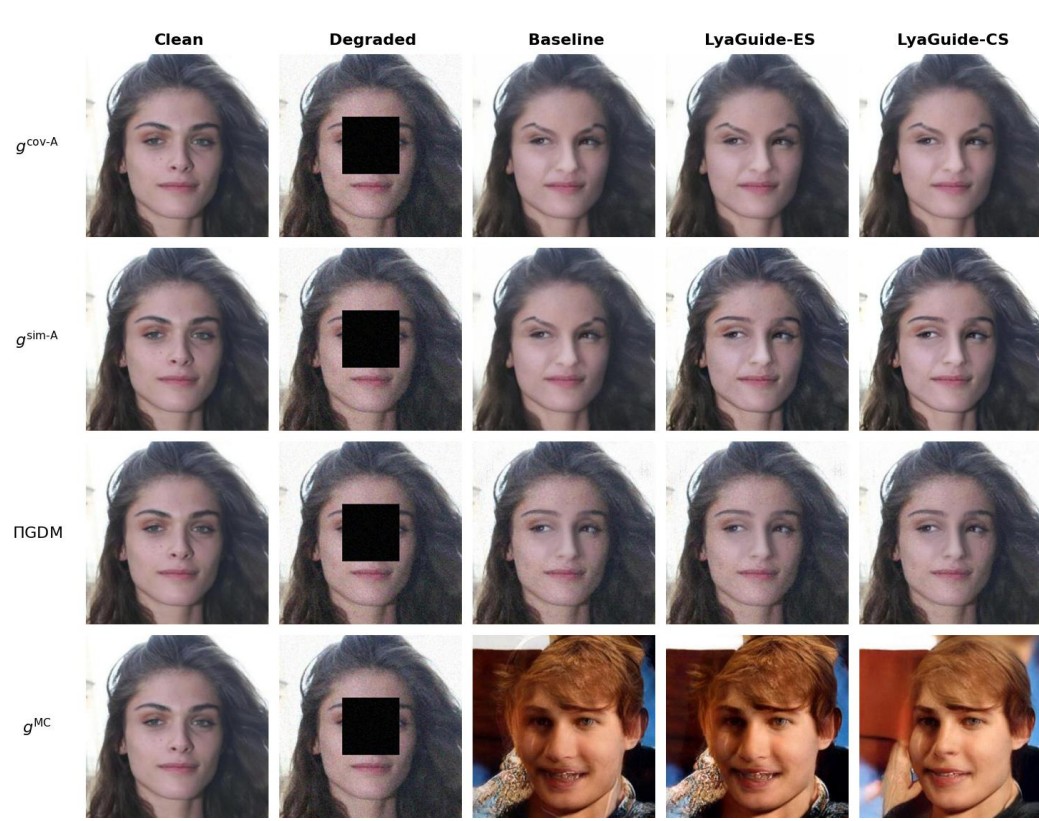

Figure 12: The visualization of the image inverse problems with the base flow matching model of conditional flow matching (CFM). Four rows show the results of four baselines with the corresponding LyaGuide results in box-inpainting task.

### A.5.3    PERFORMING LYAGUIDE TO EBM TASKS

In Prop. 3.3 we show that our LyaGuide framework works to the EBM tasks. To validate the efficacy, we evaluate the effect of applying LyaGuide on the vector field used in Energy Matching (EM) Balcerak et al. (2025), while keeping all experimental configurations identical to the original EM setup. Recall that EM sampling consists of two stages: a deterministic gradient-driven ODE phase followed by a stochastic Langevin refinement phase. In our method, we apply LyaGuide directly to the underlying vector field $-\nabla V$ used in both phases, without altering any other aspects of the sampling procedure.

Figure A.5.3 reports the results on the synthetic two-moon dataset. Under the same accuracy criterion, LyaGuide consistently requires substantially fewer integration steps and achieves a noticeable reduction in total sampling time compared to the EM baseline. Moreover, when fixing the transition time parameter $\tau^*$ in phase 1 to values relatively far from 1 (we choose 0.7 here), LyaGuide leads to trajectories that remain significantly closer to the target distribution, as reflected by faster decay of the Wasserstein-2 distance. These results demonstrate that incorporating LyaGuide into EM yields more contractive dynamics, accelerates convergence toward high-density regions, and improves sampling efficiency while preserving the original EM design.

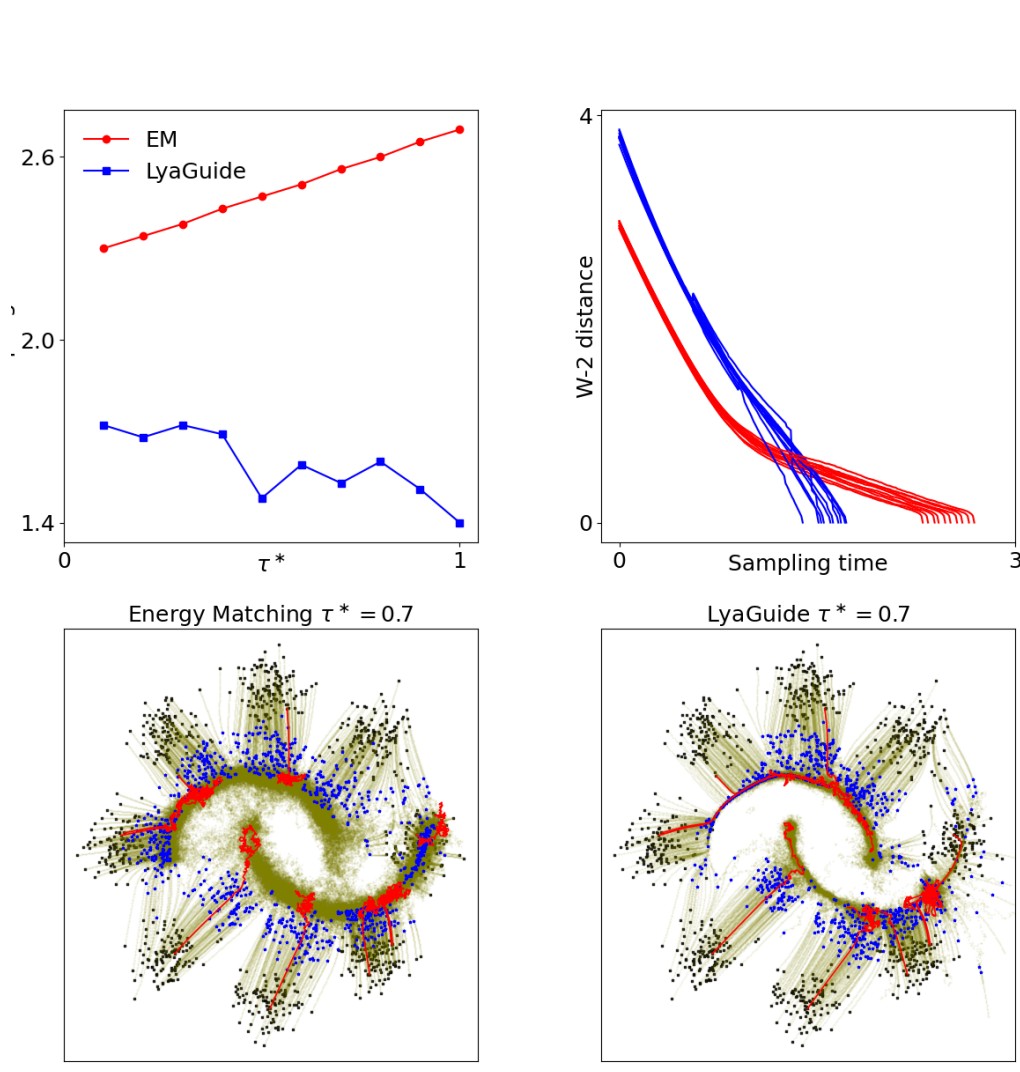

Figure 13: Upper left: Sampling time under different phase 1 duration $\tau^*$. Upper right: Wasserstein-2 distance between the sampled data and the converged data along the sampling trajectories, blue and red curves correspond to 10 trajectories under different $\tau^*$. Bottom: Diagram of the sampling process, black dots are initial samples, olive dots are converged samples, blue dots are samples at time $\tau^* = 0.7$, red curves are the sampling trajectories.

