(\boldsymbol{x}) = c_t^{\perp}(\boldsymbol{x}) + c_t^{\top}(\boldsymbol{x}), \qquad c_t^{\perp}(\boldsymbol{x}) = \alpha_t(\boldsymbol{x}) \frac{\nabla V(\boldsymbol{x})}{\|\nabla V(\boldsymbol{x})\|}, \quad \nabla V(\boldsymbol{x}) \cdot c_t^{\top}(\boldsymbol{x}) \equiv 0,$$

with the convention $\alpha_t(\boldsymbol{x}) = 0$ at critical points $\{\boldsymbol{x} : \nabla V(\boldsymbol{x}) = 0\}$. Choose

$$\alpha_t(\boldsymbol{x}) = -\frac{\boldsymbol{u}_t(\boldsymbol{x}) \cdot \nabla V(\boldsymbol{x})}{\|\nabla V(\boldsymbol{x})\|} - \delta_t(\boldsymbol{x}) \frac{V}{\|\nabla V(\boldsymbol{x})\|}, \qquad \delta_t(\boldsymbol{x}) \geq 0.$$

Then along any trajectory $\boldsymbol{x}_t$ driven by $\boldsymbol{u}_t + c_t$,

$$\frac{\mathrm{d}}{\mathrm{d}t} V(\boldsymbol{x}_t) = \nabla V(\boldsymbol{x}_t) \cdot \big(\boldsymbol{u}_t(\boldsymbol{x}_t) + c_t(\boldsymbol{x}_t)\big) = -\gamma_t(\boldsymbol{x}_t) V(\boldsymbol{x}_t),$$

so $V$ is locally Lyapunov. The tangential component $c_t^{\top}$ is then chosen to satisfy the residual of equation 10:

$$\nabla \cdot \big(p_t' c_t^{\top}\big) = p_t' \big(\boldsymbol{u}_t \cdot \nabla J + \partial_t \log Z_t\big) - \nabla \cdot \big(p_t' c_t^{\perp}\big),$$

which admits solutions $c^{\top}$ because both sides have zero integral.

**Step 3: From Lyapunov control to guided path.** Conversely, suppose there exists a control $c_t$ with

$$V = -J, \nabla V(\boldsymbol{x}) \cdot \big(\boldsymbol{u}_t(\boldsymbol{x}) + c_t(\boldsymbol{x})\big) \leq -\delta V.$$

Define

$$\phi_t(\boldsymbol{x}) := \partial_t \log Z_t + \boldsymbol{u}_t(\boldsymbol{x}) \cdot \nabla J(\boldsymbol{x}),$$

and let $\tilde{c}_t$ be *any* solution of

$$\nabla \cdot \big(p_t' \tilde{c}_t\big) = p_t' \phi_t.$$

By Step 1, the controlled field $\boldsymbol{u}_t + \tilde{c}_t$ generates $p_t'$. The family of solutions to the weighted divergence equation is affine; hence we can deform $c_t$ to $\tilde{c}_t$ by adding a field in the nullspace of the weighted divergence operator. Choosing this adjustment tangential to the level sets of $J$ or $V$ (i.e. orthogonal to $\nabla J$ or $\nabla V$) preserves the Lyapunov inequality while matching the required divergence. Thus both the Lyapunov condition and equation 10 can be enforced simultaneously.