# OpenReview forum: "Lyapunov Guidance: Stabilizing Generative Flows with One-Line Code"
_ICLR.cc/2026/Conference — Submitted to ICLR 2026_

### Official Review · Reviewer_NCg4 · 2025-10-29

**Soundness:** 3
**Presentation:** 2
**Contribution:** 3
**Rating:** 6
**Confidence:** 3

**Summary:**

This paper proposes a unified framework, Lyapunov Guidance for flow matching (LyaGuide), which reformulates the guidance in flow matching as a Lyapunov control problem. Based on this framework, the paper designs a pseudo-projection operator with a closed-form expression that strictly satisfies the Lyapunov condition. The experiments demonstrate the effectiveness of the proposed method.

**Strengths:**

- This paper offers a new framework or perspective for the flow matching guidance problem.
- Based on the stability theory in the controlled system, the paper proposes a projection operation for more accurate guidance.
- The method can be applied to broad scenarios: both explicit and implicit prior knowledge.

**Weaknesses:**

- l146: What does this sentence 'c is a control term derived from V' mean? Or, what relationship is 'derived from'? This is not clear here, although later there are detailed explanations. It will be better to explain it here.
- The caption of Fig.2 is unclear. Maybe change it to 'Illustration of the pseudo projection $\pi$ and exact projection $\pi^*$? Additionally, the meaning of the points with different colors in this figure is not clear.
- How does the sampling step of flow matching influence the performance of LyaGuide?

**Questions:**

- l130: Why is designing the stabilizing controller u(x) a major problem in cybernetics field? Or, why is the stable u(x) better? It needs a more intuitive explanation for easier understanding.
- l350: Why should V be locally convex around task-relevant regions? Proposition 3.1 does not include such assumptions. Also, why does the importance sampling weight promote the convexity around high-score samples? And if the locally convexity is needed, what about section 5.1? Can V be locally convex in this setting?
- l366: what is $g_ϕ$? This may need a brief introduction.

---

> ### Author Response · Authors · 2025-11-22
> **Response to the Reviewer NCg4's comments (1/2)**
>
> We would like to thank the reviewer for the comments and valuable suggestions. The valuable comments help us to improve the quality of the manuscript. We address the concerns of the reviewer one by one.
>
> **W1**: l146: What does this sentence 'c is a control term derived from V' mean? Or, what relationship is 'derived from'? This is not clear here, although later there are detailed explanations. It will be better to explain it here..
>
> **Response**: Many thanks for your comment. Here we want to state that the guidance term $c$ should be derived from the conditional distribution $e^{-J(x)}$ such that the guided flow can sample the distribution $\frac{1}{Z}p_1(x)e^{-J(x)}$. The construction process of $c$ can be equated to a control problem wherein the control term $c$ is devised to satisfy a Lyapunov condition $V$, where $V$ is related to the energy function $J$. We modify our presentation in the uploaded revision.
>
> **W2**: The caption of Fig.2 is unclear. Maybe change it to 'Illustration of the pseudo projection $\pi$  and exact projection $\pi^\ast$? Additionally, the meaning of the points with different colors in this figure is not clear.
>
> **Response**: Thanks for your careful reading and constructive comment. We have revised the caption of Fig.~2 as follows:
> 'Illustration of the pseudo projection $\pi$  and exact projection $\pi^\ast$. Here the grey dot is the candidate control $c$, the purple dot is the projected element $\pi^\ast(c_t)$ of $c$ in the target space $\mathcal{U}(V)$, and the yellow dot is the pseudo projected element $\pi(c_t)$.' Please refer to the uploaded revision.
>
> **W3**: How does the sampling step of flow matching influence the performance of LyaGuide?
>
> **Response**: Many thanks for your valuable comment. To investigate how the sampling time horizon and the number of flow-matching steps affect LyaGuide, we conducted an early-inference-termination experiment. Specifically, instead of integrating the flow matching dynamics up to $t=1$, we stop the ODE at intermediate times and measure the Wasserstein-2 distance between the partially evolved samples and the target distribution. The results are shown in Figure 5. Across all eight methods, LyaGuide consistently achieves lower W-2 distances at every inference time, indicating that its trajectories contract toward the target distribution significantly faster than those of the original guidance fields. In particular, the gap between the baseline and LyaGuide is already substantial at early inference times (e.g., $t \approx 0.4$), demonstrating that the Lyapunov-based correction accelerates convergence even when the flow is only partially executed.
> These results further show that LyaGuide reduces the dependence of flow matching on fine integration granularity: even with fewer integration steps or shorter time horizons, the guided dynamics maintain stability and approach the target distribution more rapidly than the unguided flows. This suggests that the Lyapunov structure not only improves final-time sample quality but also enhances transient convergence efficiency. The results are also consistent with theoretical intuition, since we know that the Lyapunov control steers the controlled flow towards the minimizers of Lyapunov function with an exponential convergence rate, so it naturally accelerates the sampling process of flow matching. We have added this analysis and discussion to the revised manuscript to address the reviewer’s concern.
>
> **Q1**: l130: Why is designing the stabilizing controller u(x) a major problem in cybernetics field? Or, why is the stable u(x) better? It needs a more intuitive explanation for easier understanding.
> Response: Many thanks for your careful reading and helpful comment. Stabilization is a fundamental objective that appears across many areas of cybernetics and control theory. Numerous control problems can be reformulated as achieving stability around a desired state. Examples include: tracking a target trajectory in vehicle control [R1]; synchronizing neuronal populations in the treatment of neurological disorders such as Alzheimer’s disease via deep brain stimulation (DBS)[R2]; regulating mRNA expression levels in synthetic biology, which amounts to steering a gene regulatory network from one equilibrium to another [R3]; and enhancing the classification performance of neural ode [R4]. In all these settings, a stable controller $u(x)$ is essential for achieving the intended outcome.
>
> For generative models such as flow matching, an analogous notion of stability arises: a stable guidance field helps ensure that the guided flow converges reliably toward the target data distribution. As discussed in our response to **W3**, enforcing stability not only improves convergence but can also significantly reduce inference time. This is precisely the motivation for exploring Lyapunov-based guidance in our work.

---

> > ### Author Response · Authors · 2025-11-22
> > **Response to the Reviewer NCg4's comments (2/2)**
> >
> > **Q2**: l350: Why should V be locally convex around task-relevant regions? Proposition 3.1 does not include such assumptions. Also, why does the importance of sampling weight promote the convexity around high-score samples? And if the local convexity is needed, what about section 5.1? Can V be locally convex in this setting?
> > Response: Many thanks for your careful reading and helpful comment. You are absolutely right that Proposition 3.1 does not assume any form of local convexity of $V$, and our previous presentation in l350 was misleading. Our method does not require $
> > V$ to be locally convex. What we need is much weaker: during learning, $V$ should correctly identify task-relevant local minima (i.e., low-energy / high-density regions). The importance-weighting scheme is used to bias the learner toward accurately fitting low-energy samples, where the Lyapunov guidance is most influential. We have revised the text to reflect this more precise meaning. Regarding Section 5.1, no convexity assumption is imposed there either. We thank the reviewer for catching this issue and have corrected the wording accordingly.
> >
> > **Q3**:l366: what is $g_\phi$? This may need a brief introduction.
> >
> > **Response**: Many thanks for your comment.  The term $g_\phi$ follows the formulation in [R5], and is trained using the Guidance Matching Loss introduced in this paper. The idea is to first learn the normalizing constant of the conditional distribution $Z_\{\phi}\approx Z_t(x_t,t)=\int e^{-J(x_t)}p(z|x_t)$, where $z$ is the conditioner. Then, according to the theory proposed in [R5], guided flow should satisfy $g_t=\int F(e^{-J},Z_t)u_t(x_t|z)p_t(z|x_t) dz$. With the learned $Z_{\phi}$, another neural network $g_\phi$ can be trained to enforce the equation constraint under the loss function $L=\mathbb{E}\_{x,z,t}[g_{\phi}(x,t)-F(e^{-J}(x),Z_t(x,t))u_t(x_t|z)]$. Here $F$ is some known functional over $e^{-J}$ and $Z_t$.
> >
> > Finally, we would like to thank the reviewer again for his/her time and positive feedback on the paper. The revised paper is improved a lot from the theoretical and experimental aspects. We hope that the reviewer will be satisfied with the responses and the supplemented results as well, and then consider revising the assessment in support of the revised paper. We may make further improvements according to your feedback.
> >
> > **References**
> >
> > [R1] Dawson, C., Gao, S., & Fan, C. (2023). Safe control with learned certificates: A survey of neural lyapunov, barrier, and contraction methods for robotics and control. IEEE Transactions on Robotics, 39(3), 1749-1767.
> >
> > [R2] Yang, L., Zhang, J., Zhou, S., & Lin, W. (2025). Advancements in Mathematical Approaches for Deciphering Deep Brain Stimulation: A Systematic Review. CSIAM Trans. Life Sci, 1(1), 93-133.
> >
> > [R3] Wang, L. Z., Su, R. Q., Huang, Z. G., Wang, X., Wang, W. X., Grebogi, C., & Lai, Y. C. (2016). A geometrical approach to control and controllability of nonlinear dynamical networks. Nature communications, 7(1), 11323.
> >
> > [R4] Rodriguez, I. D. J., Ames, A., & Yue, Y. (2022, June). Lyanet: A lyapunov framework for training neural odes. In International conference on machine learning (pp. 18687-18703). PMLR.
> >
> > [R5] Feng, R., Yu, C., Deng, W., Hu, P., & Wu, T. (2025). On the guidance of flow matching. arXiv preprint arXiv:2502.02150.

---

> > > ### Comment · Reviewer_NCg4 · 2025-11-26
> > >
> > > Thanks for the detailed response from the authors! They have addressed my concerns, and I will maintain my score.

---

> ### Author Response · Authors · 2025-11-27
>
> Thanks again for your great effort in reviewing this paper and the suggestions are so valuable that we would like to polish the algorithm in future work. We wonder that whether you could raise the score if all the concerns have been addressed?

---

### Official Review · Reviewer_wGTM · 2025-10-30

**Soundness:** 2
**Presentation:** 3
**Contribution:** 4
**Rating:** 4
**Confidence:** 2

**Summary:**

This paper introduces LyaGuide, a Lyapunov function-based method to enhance guidance in flow matching generative models, addressing inefficiencies in adapting pre-trained models to new tasks (e.g., inverse problems like image restoration). Drawing from control theory, it provides a unified, stable, and theoretically guaranteed approach that integrates with existing methods, requires minimal code changes, and improves reliability for applications in drug design, image editing, and beyond.

**Strengths:**

1. It's an interesting (and the first, to the knowledge of the reviewer) attempt to reformulate the guidance problem in generative models as a control stability problem, and studying it with Lyapunov functions.
2. The proposed theoretical framework includes different guidance techniques, including EBM guidance (since the guidance is the gradient of a time-invariant potential) and posterior sampling in diffusion models and flow matching.

**Weaknesses:**

1. The presentation can be improved. For example, a more detailed discussion and intuitive understanding of the equivalence between guidance and Lyapunov stability in the main text would be beneficial.
2. The experimental evaluation is relatively limited. E.g., only flow matching on an image inverse problem is evaluated, whereas the authors claim that the framework is generally applicable to EBM and various tasks. Surely, the contribution of this work is largely theoretical, but the soundness would be improved with more empirical evidence.

**Questions:**

1. In my understanding, Theorem A.2.1 is central to the contributions. However, how the equivalence between Lyapunov stability and guidance is established can be presented more clearly. In my understanding, the proof shows that a locally Lyapunov stable control can be "deformed" to the desired guidance control, but it does not show how it can be deformed. Lastly, it is not entirely clear how the proposed method, after the control is projected to be Lyapunov stable, deforms the stable control to match the desired guidance control.
2. What is causing the Lypapunov guidance results to still be different from ground truth?

I am willing to raise the score if the questions are answered and the concerns are addressed.

Minor:
Line 227 guidnace -> guidance

---

> ### Author Response · Authors · 2025-11-22
> **Response to the Reviewer wGTM's comment (1/2)**
>
> We thank the reviewer for the overall positive feedback and the valuable comments. We have revised the manuscript according to the valuable comments. We hope the reviewer could find the efforts and the improvements we made not only from the theoretical aspect and the experimental aspect. For the individual comments, we are going to respond to them one by one.
>
> **W1**: The presentation can be improved. For example, a more detailed discussion and intuitive understanding of the equivalence between guidance and Lyapunov stability in the main text would be beneficial.
>
> **Response**: Thanks for your thoughtful suggestion. We agree that the intuition behind the equivalence between guidance and Lyapunov stability can be further clarified. In the revision, we have expanded the main-text discussion to make this correspondence more accessible. Our Theorem 3.2 show that Lyapunov-stable controls (those satisfying the Lyapunov condition in Prop. 3.1) and guidance-compatible controls (those satisfying the weighted divergence equation 10 in the Appendix) are mutually convertible. Converting a Lyapunov-stable control into a guidance term requires solving a coupled PDE system, as detailed in Appendix A.2.1 and the answers to your question Q1. This direction is generally expensive and difficult. The key insight of our method is that the reverse direction of deforming from a guidance term to a Lyapunov-stable control, is far more tractable. Once a guidance term is given (as in standard FM/SGM guidance methods), we can enforce Lyapunov stability through the projection operator in Theorem 4.1, which has analytical expression and can be implemented efficiently. This projection ensures that the resulting vector field preserves both the desired guidance property and the Lyapunov decrease condition. We have revised the main text to highlight this conceptual symmetry and to provide a more intuitive explanation of why the projection-based direction is computationally efficient and central to the proposed LyaGuide approach.
>
> **W2**: The experimental evaluation is relatively limited. E.g., only flow matching on an image inverse problem is evaluated, whereas the authors claim that the framework is generally applicable to EBM and various tasks. Surely, the contribution of this work is largely theoretical, but the soundness would be improved with more empirical evidence.
>
> **Response**: In our experiments, we have already included CEG (Contrastive Energy Guidance), which is an energy-based generative method and belongs to the family of EBM variants. In addition, we are implementing further numerical comparisons on RL tasks and more EBM-based baselines. Due to limited GPU resources, these results are still being computed, and we expect to release them next week. We will kindly notify you once the updated results are posted.
>
> **Q1**: In my understanding, Theorem A.2.1 is central to the contributions. However, how the equivalence between Lyapunov stability and guidance is established can be presented more clearly. In my understanding, the proof shows that a locally Lyapunov stable control can be "deformed" to the desired guidance control, but it does not show how it can be deformed. Lastly, it is not entirely clear how the proposed method, after the control is projected to be Lyapunov stable, deforms the stable control to match the desired guidance control.
>
> **Response**: Thanks for your valuable comment. We clarify how the deformation from a Lyapunov-stable control to the desired guidance control is constructed, and why our projection based method matches the desired guidance control as follows.
>
> ```(1) Explicit deformation: solving auxiliary PDEs.``` Given a Lyapunov-stable control $c_t$, we correct it to satisfy the divergence equation by solving the coupled PDE system $\nabla V⋅w_t=0, \forall x\in O(x^\ast,\epsilon)$, $\nabla\cdot(p'_tw_t)=p'_t(u_t\cdot\nabla J + \partial_t\log Z_t)$.The first equation enforces tangency to the Lyapunov level sets, ensuring that the corrected control preserves the Lyapunov decrease condition. The second equation corrects the divergence defect. We have verified that the second weighted divergence equation is solvable in our revised manuscript A.2.1. Thus the deformation $\tilde{c}_t=c_t+w_t$ is explicitly obtained by solving this PDE system.
> ```(2) What's the role of projection?```
> We note that our Theorem establishes the coexistence of Lyapunov-stable and guidance-compatible controls. Although we show that one can start from a Lyapunov-stable control and enforce the divergence equation by solving the PDEs above, this direction is generally more difficult because it requires solving the auxiliary PDEs. In contrast, starting from a guidance term that already satisfies the divergence constraint is straightforward, and Lyapunov stability can be imposed by an intractable projection (Theorem 4.1) without solving any PDE.
> We have clarified these points in the revised Appendix A.2.1.

---

> ### Author Response · Authors · 2025-11-22
> **Response to the Reviewer wGTM's comment (2/2)**
>
> **Q2**:  What is causing the Lypapunov guidance results to still be different from ground truth?
>
> **Response**: Many thanks for your interesting comment. The reasons are two-fold. First, in Scenario 1 where a Lyapunov function is known a priori, the Lyapunov condition ensures that samples generated by the guided flow converge exponentially fast to the ground-truth distribution, at a rate $e^{-\delta t}$. In practice, however, the flow is simulated only over a finite interval $[0,1]$. If we extend the simulation horizon $T$, or equivalently choose a larger Lyapunov exponent $\delta$, the guided samples converge much closer to the ground truth, as shown in Fig. 10 of the revised manuscript. Furthermore, supplemented Fig. 5 demonstrates that LyaGuide achieves substantially faster convergence to the ground truth than all baseline methods, indicating that early inference with LyaGuide is indeed feasible.
> Second, in Scenario 2 where the Lyapunov function is not known and must be learned from data, the learned $V$ inevitably introduces approximation error relative to the true underlying potential. This approximation error propagates into the subsequent LyaGuide updates, which explains why the final guided samples may deviate from the ground-truth distribution.
>
>
> We thank the reviewer again for your valuable comments. We do believe that the quality of the revised paper has been improved exceptionally from the experimental and theoretical aspects thanks to the reviewers’ comments and suggestions. Hopefully, the responses and the revised paper have sufficiently addressed the main concerns and then the reviewer will reconsider the assessment in support of the revised paper. We are looking forward to your feedback to make further improvements of the paper.

---

> > ### Comment · Reviewer_wGTM · 2025-11-22
> >
> > Thank you for the detailed response. I believe the clarity of the revised manuscript has been improved and thus increased my score.
> >
> > However, regarding Theorem 3.2, I find the expression "performing conditional sampling is equivalent to finding the controller that satisfies the local Lyapunov condition" confusing to some extent. According to my understanding, a local Lyapunov stable control is neither a sufficient nor a necessary condition of a guidance that achieves conditional sampling. Therefore, I recommend the authors to rephrase the sentence to, e.g., "there exists a control that is both locally Lyapunov stable and generating the guided probability path", to avoid potential misunderstanding.
> >
> > Besides, I have an additional question about Theorem 4.1. After projection, does the new control $\pi(c_t, U)$ still preserve the weighted continuity equation? If not, a clarification here would be beneficial, since the simultaneous satisfaction of local Lyapunov stability and weighted continuity equation (i.e., an exact guidance) seems to be closely related to the theorems presented earlier in the manuscript.

---

> > > ### Author Response · Authors · 2025-11-24
> > > **Response to Reviewer's reply**
> > >
> > > Thank you very much for the constructive follow-up comment and for increasing your score. We are glad that the clarifications in our rebuttal improved the readability of the manuscript. For your further comments, we reply them as follows.
> > >
> > > ```On the phrasing of Theorem 3.2.```
> > > We agree that the original presentation ``performing conditional sampling is equivalent to finding the controller that satisfies the local Lyapunov condition.’’ in Theorem 3.2 is not clear. And we rephrase our Theorem according to your suggestion
> > >
> > > ```
> > > There exists a control that is both locally Lyapunov stable and generates the guided probability path, which therefore accelerates the sampling process of guided flows.
> > > ```
> > > This phrasing avoids any possible misunderstanding and more faithfully captures the meaning of Theorem 3.2, and the supplemented results in Fig. 5 validate the claim of the Theorem 3.2.
> > >
> > > ```
> > > On whether the projected control in Theorem 4.1 preserves the weighted continuity equation.
> > > ```
> > > This is an excellent question, and we appreciate the opportunity to clarify it. At present, the pseudo-projection in Theorem 4.1 is designed solely to enforce the Lyapunov decrease condition, and, in general, it does not preserve the weighted continuity equation (Eq. 10 in the appendix). In this sense, our construction is partially heuristic: the aim is to obtain a guidance term that is provably Lyapunov-stable, thereby accelerating sampling and improving robustness, even though the divergence constraint may no longer be exactly satisfied after projection. The acceleration and robustness effects are reflected in Fig. 5 and Table 2 in the revised manuscript.
> > >
> > > Providing a rigorous correction operator that simultaneously guarantees both local Lyapunov stability and the weighted continuity equation remains an interesting and nontrivial direction for future research. We have added a discussion of this limitation and its implications in the revised manuscript; please refer to Appendix A.2.6 for details.
> > >
> > > Once again, we sincerely thank the reviewer for the thoughtful feedback and the encouraging update to the evaluation score. We will incorporate the suggested refinements into the revised version to further improve clarity and precision.

---

> > > > ### Author Response · Authors · 2025-11-25
> > > > **Update of the response to Reviewer wGTM**
> > > >
> > > > We thank the reviewer for the valuable suggestion regarding the applicability of our framework to EBM tasks. In Proposition 3.3, we theoretically establish that LyaGuide naturally extends to energy-based models. To further validate this claim, we conducted additional experiments by integrating LyaGuide into the Energy Matching (EM) method, while keeping all experimental configurations identical to the original EM setup. EM involves two stages: a deterministic ODE phase driven by $-\nabla V$ and a subsequent Langevin refinement phase. In our evaluation, LyaGuide is applied directly to the same drift vector field $-\nabla V$ in both phases, without modifying any other components of EM.
> > > >
> > > > The results in supplemented Fig. 13 show that, under an identical accuracy threshold, LyaGuide consistently requires substantially fewer integration steps and yields a clear reduction in total sampling time compared with the EM baseline. Moreover, when fixing the phase-1 transition parameter $\tau^\ast$ to values farther from $1$ (we set $\tau^\ast = 0.7$), LyaGuide produces trajectories that remain significantly closer to the target distribution, as evidenced by a faster decay of the Wasserstein-2 distance. These findings demonstrate that incorporating LyaGuide into EM induces more contractive dynamics, accelerates convergence toward high-density regions, and improves sampling efficiency while fully preserving the original EM design.

---

### Official Review · Reviewer_qnFP · 2025-10-31

**Soundness:** 3
**Presentation:** 3
**Contribution:** 2
**Rating:** 4
**Confidence:** 3

**Summary:**

This paper introduces LyaGuide, a framework that unifies various flow-matching guidance methods (e.g., classifier guidance, energy-based guidance, reward guidance) under the theoretical perspective of Lyapunov control. The authors propose interpreting the energy function in generative models as a Lyapunov function and the guidance term as a stabilizing control input. To ensure stability, they introduce a pseudo projection operator that enforces the Lyapunov condition in closed form, claiming compatibility with existing methods and implementation simplicity (“one-line code”). Experiments on synthetic datasets and image inverse problems (inpainting, super-resolution, deblurring) show improvements in stability and performance over baseline guidance methods.

**Strengths:**

- Unifies diverse guidance techniques in generative modeling under Lyapunov control theory, offering a new theoretical lens.

- The pseudo-projection operator provides a lightweight, closed-form correction that can be easily integrated into existing models.

- The framework can wrap around multiple existing guidance methods, making it versatile.

- Synthetic and image inverse experiments show modest but consistent improvements in stability and quality.

**Weaknesses:**

- Several stability conditions are unverified or incorrectly generalized from local to global settings.

- Quantitative improvements (e.g., in Fig. 3 and Table 1) are marginal, and visual results show minor differences; the proposed method mainly stabilizes trajectories rather than improving fidelity substantially.

- Fig. 3’s third row labeling appears incorrect, and the “toy example” results do not clearly demonstrate meaningful gains.

- Lyapunov functions ensure local minima convergence, not global optimization, limiting practical utility.

- The phrase “rigorous stability guarantees” is misleading given the local and heuristic nature of the verification.

**Questions:**

- How can one verify that the proposed $V(x)$ satisfies Lyapunov’s positive definiteness and negative derivative conditions in practice?

- What is the basin of attraction or stable manifold for convergence, and how can users determine if $x_0$ lies within it?

- Does the pseudo-projection operator guarantee convergence when $V(x)$ is non-convex or multimodal?

- How sensitive is the method to scaling parameters $(δ, k)$ in the projection and control terms?

- Line 185 has a typo "exists a a continuously ".

---

> ### Author Response · Authors · 2025-11-23
> **Response to the Reviewer qnFP's comments (1/3)**
>
> We thank you for your valuable comments, and helpful suggestions on the proposed methods. For the minor/major points in this work, we respond to them one by one. Hopefully, the following answers could help you understand our work better.
>
> **W1**: Several stability conditions are unverified or incorrectly generalized from local to global settings.
>
> **Response**: Thanks for your valuable comments. We agree that the original manuscript could be interpreted as implying global stability. In the revised manuscript, we stress that all our theoretical results, including Proposition 3.1 and Theorem 4.1, only require **local Lyapunov stability near task-relevant minima**, not global stability. Both the positive definiteness condition and Lyapunov inequality condition are required to satisfied in a small neighborhood of local minima of $V$.
>
> **W2**: Quantitative improvements (e.g., in Fig. 3 and Table 1) are marginal, and visual results show minor differences; the proposed method mainly stabilizes trajectories rather than improving fidelity substantially.
>
> **Response**: Thanks for your valuable comments. We agree that the improvement in Table 1 is relatively small. However, as shown in Fig. 3, the samples generated under LyaGuide visibly converge more closely toward the ground-truth distribution compared to their unguided counterparts.
> Beyond final-time sample fidelity, we further conducted early-termination experiments to quantify how LyaGuide affects inference dynamics. Instead of integrating the flow-matching ODE up to $t=1$, we stop the trajectory at intermediate times and compute the Wasserstein-2 distance between the partially evolved samples and the target distribution. The results (Fig. 5) show a clear and consistent trend:
> - **Across all eight baselines, LyaGuide achieves strictly lower W-2 distance at every inference time.**
> - **The gap becomes substantial even in early stages** (e.g., around $t \approx 0.4$), well before the end of the flow.
> - This indicates that the LyaGuide-corrected trajectories contract toward the target distribution **significantly faster** than those guided by the original vector fields.
>
> These findings highlight a key advantage not captured by final-time metrics alone:
> **LyaGuide reduces the dependence of flow matching on long sampling horizons**.
>  Even with fewer integration steps or truncated time horizons, the guided dynamics remain stable and approach the target distribution much more rapidly.
> This behavior is fully aligned with our theoretical intuition. The Lyapunov correction explicitly enforces a contracting structure around task-relevant regions, yielding **exponential-type convergence toward the minima of $V$**. As a result, LyaGuide not only stabilizes trajectories but also **accelerates the sampling process**, improving transient convergence efficiency in addition to modest steady-state improvements.
>
> **W3**: Fig. 3’s third row labeling appears incorrect, and the “toy example” results do not clearly demonstrate meaningful gains.
>
> **Response**: Thanks for your valuable comments. We have corrected the labeling issue in the third row of Fig. 3. To more clearly quantify the benefit of LyaGuide beyond visual inspection, we additionally evaluated its performance using the Wasserstein-2 (W-2) distance. As shown in the new results (Fig. 5 and Table 2), LyaGuide achieves **significantly lower W-2 distance across all inference times**, indicating that its trajectories contract toward the target distribution much faster than the original guidance fields. This demonstrates that LyaGuide not only reduces the final W-2 error but also **accelerates sampling**, yielding more accurate intermediate states even when the ODE integration is truncated.
> These quantitative results complement the original results and provide strong evidence that LyaGuide improves both convergence quality and inference efficiency.
>
> **W4**: Lyapunov functions ensure local minima convergence, not global optimization, limiting practical utility.
>
> **Response**: Thanks for your valuable comments. We would like to clarify the notion of “global optimization” in this context. In control theory, the relevant distinction is between **global stability** (a single global equilibrium acting as an attractor) and **local stability** (multiple equilibria, each with its own local basin). Our work follows the latter, because the target distribution in generative modeling naturally contains multiple modes, each corresponding to a different local minimizer of the potential $V$.
> In such multi-modal settings, global stability is neither realistic nor desirable: enforcing a single global attractor would collapse all modes. Hence we focus on local stability around each mode, ensuring that trajectories contract when they enter the basin of the corresponding data mode. This is exactly what LyaGuide leverages. The Lyapunov function is used to guarantee local contraction toward the correct mode, not to solve a global optimization problem.

---

> ### Author Response · Authors · 2025-11-23
> **Response to the Reviewer qnFP's comments (2/3)**
>
> In addition, **if the target distribution contains only a single minimizer**, then our pseudo-projection mechanism automatically enforces global stability. We have therefore added a Theorem in the revised manuscript to explicitly state and prove this global guarantee in the single-equilibrium case.
> Therefore, the use of a local Lyapunov function is not a limitation of our method but a direct reflection of the structure of the underlying generative problem. Moreover, as shown in Fig. 5, the resulting local contraction significantly accelerates convergence and reduces W-2 distance across all inference times.
> We have also supplemented the above discussion into the revised manuscript. We thank the reviewer again for the valuable comment.
>
> **W5**: The phrase “rigorous stability guarantees” is misleading given the local and heuristic nature of the verification.
>
> **Response**: Thanks for your valuable comments. We have revised the manuscript to clarify the meaning of “rigorous stability guarantees’’ and to avoid any possible misinterpretation. The rigor in our guarantees refers specifically to the **local Lyapunov stability** formalized in Proposition 3.1. Theorem 4.1 provides a mathematically exact guarantee that, after applying our pseudo-projection operator, the corrected guidance field **strictly satisfies the Lyapunov decrease condition on the entire neighborhood $O(x^\ast,\varepsilon)$** around each target minimizer $x^\ast$. This verification is fully theoretical and does not rely on heuristic assumptions.
>
> Furthermore, when the target distribution contains a **single global minimizer**, the same pseudo-projection mechanism yields a **global Lyapunov stability guarantee**, and we have added a theorem in the revised version to explicitly state this global result.
> To distinguish clearly between heuristic and non-heuristic components:
> - The empirical training of the candidate potential $V_{\theta_V}$ is heuristic, since a finite-sample Lyapunov loss cannot ensure correctness across the entire continuous region.
> - The pseudo-projection step is **not heuristic**; it converts the learned vector field into one that provably satisfies the Lyapunov condition everywhere within the target region, eliminating reliance on sample-based verification.
> We have updated the presentation accordingly to ensure that “rigorous stability guarantees’’ refers precisely to these mathematically verified properties of the pseudo-projection method.
>
> **Q1**: How can one verify that the proposed $V(x)$ satisfies Lyapunov’s positive definiteness and negative derivative conditions in practice?
>
> **Response**: Thanks for your valuable comments. We clarify how Lyapunov’s positive definiteness and negative-derivative conditions are handled in our framework as follows:
>
> ```Positive definiteness of $V(x)$.```
> Our theory does not rely on a single fixed Lyapunov function, but on an entire equivalence class of Lyapunov candidates obtained through affine transformations. In particular, if a candidate $V(x)$ is not strictly positive definite, we can always shift it by a sufficiently large constant $k>0$ and form $\tilde V(x) = V(x) + k$, which preserves all derivative properties while ensuring $\tilde V(x) > 0$ on the domain of interest. Then we apply the subsequent guidance procedures based on $\tilde V$,
> Moreover, near a local minimizer $x^\ast$, any continuous $V(x)$ with $V(x^\ast)=0$ typically satisfies a local lower bound of the form $V(x) \ge c\,||x - x^\ast||^p,$
> which establishes the required local positive definiteness.
>
> ``` Negative-derivative condition.```
> As discussed in our response to **W5**, the empirical loss used to train $V_{\theta_V}$ provides only sample-level verification. The rigorous guarantee comes from Theorem 4.1, where the pseudo projection maps any guidance field into one that provably satisfies $\nabla V(x)\cdot (u_t(x) + g_t(x)) \le -\delta(x)V(x),$ throughout the entire local neighborhood $O(x^\ast,\varepsilon)$.
> Thus, the derivative condition is enforced analytically rather than heuristically.
>
> In summary, positive definiteness is ensured by affine adjustment of $V$ and its local minima structure, while the negative-derivative condition is rigorously guaranteed by the pseudo-projection formalized in Theorem 4.1.
>
> **Q2**: What is the basin of attraction or stable manifold for convergence, and how can users determine if $x_0$ lies within it?
>
> **Response**: Thanks for your valuable comments. In the setting of Lyapunov-based stability with multiple local minima, the region of attraction (ROA) of each minimizer $x^\ast$ is the set of points whose trajectories converge to that minimizer under the guided flow. The boundaries between ROAs are determined by the stable manifolds of saddle points or local maxima of $V$. These boundaries partition the entire state space into disjoint ROAs corresponding to different local minima.

---

> > ### Author Response · Authors · 2025-11-23
> > **Response to the Reviewer qnFP's comments (3/3)**
> >
> > Our pseudo-projection operator guarantees that, within any given ROA, all trajectories satisfy the Lyapunov decrease condition and therefore converge to the corresponding local minimum. For any initial state $x_0$ lying inside an ROA, the role of LyaGuide is to accelerate convergence toward the associated minimizer without changing the underlying attraction structure.
> >
> > The only case where the guarantee does not apply is when $x_0$ lies exactly on the ROA boundary, where the vector field is not well-defined in terms of Lyapunov decrease. However, ROA boundaries have Lebesgue measure zero in the continuous state space, meaning that for any absolutely continuous initialization, such as the Gaussian initial distributions commonly used in flow matching, the probability of sampling a point exactly on the boundary is zero. Thus, this edge case does not affect practical applicability.
> >
> > **Q3**: Does the pseudo-projection operator guarantee convergence when $V(x)$ is non-convex or multimodal?
> >
> > **Response**: Many thanks for your valuable comment. The proposed pseudo-projection in Theorem 4.1 guarantees the guided flow converges towards the local minima points of $V(x)$ that in the same ROA of initial $x_0$. The conclusion of Theorem 4.1 holds without requiring $V$ to be convex, nor the landscape to be unimodal. On the contrary, as we answered to the above weaknesses and questions, in our paper we mainly focus on non-convex and multimodal $V$ that has **multiple local minima**, each with its own region of attraction (ROA). The pseudo projection operates **within each ROA**: for any initial point $x_0$ inside the basin of a given minimizer, the corrected guidance field guarantees contraction of $V$ and thus convergence to that local minimum. In other words, LyaGuide preserves the multi-modal structure and provides Lyapunov-based convergence in each basin, rather than collapsing everything to a single mode.
> >
> > As discussed in our response to **W4**, in the special case where the target distribution has a single minimizer, the same pseudo-projection reduces to a global Lyapunov guarantee.
> >
> > **Q4**: How sensitive is the method to scaling parameters $(\delta,k)$ in the projection and control terms?
> >
> > **Response**: Thanks for your careful reading and helpful comment. We systematically investigate the influence of the parameters $(\delta,k)$ in an ablation study on synthetic dataset. The results are summarised in Table 2 and Figure 10. Based on these new results, we address the reviewer's concern as follows:
> >
> > ```Sensitivity with respect to $\delta$.``` The parameter $\delta$ appears both in the Lyapunov convergence rate in Prop 3.1 and in the explicit projection term in Theorem 4.1. From Figure 10, we observe that:
> > - Overall, the method is robust to $\delta$ across a broad range of values.
> > - As $\delta$ increases, the generated samples concentrate more strongly near the minima of the Lyapunov function V.
> > - This stronger contraction improves convergence but also reduces the probability of visiting low-density regions, thereby decreasing exploration capability.
> > In other words, large $\delta$ induces over-contraction, which may excessively bias sampling towards high-density areas.
> > Therefore, $\delta$ can be tuned according to the task requirements. The results also show excessively large $\delta$ may lead LyaGuide to underperform compared to the original method. Based on our empirical findings, a practical recommendation is to choose $\delta\in(0,1]$.
> >
> > ```Sensitivity with respect to $k$ (gradient-based guidance only)```.
> > The control coefficient $k$ only affects gradient-based guidance methods (e.g., energy or score-based guidance), appears in the initial guidance term $-k\nabla V$ without LyaGuide. So we evaluate $k$ exclusively on the gradient-based variants (Table 2).
> > The results show that:
> > - Increasing $k$ generally improves sample quality, as stronger control in the Lyapunov direction facilitates more effective contraction of the gradient flow.
> > - The benefit of larger $k$ persists across different $\delta$ values, although extreme $\delta$ can diminish this improvement.
> > This confirms that $k$ acts as a refinement mechanism for stabilising gradient-based methods, rather than a global sensitivity parameter.
> >
> > We thank the reviewer again for such a valuable comment. These new ablation studies have been added to the manuscript to address the reviewer’s concerns.
> >
> > Q5: Line 185 has a typo "exists a a continuously ".
> > Response: Thanks for your careful reading. We have corrected this typo in the revision.
> >
> > Finally, we would like to thank the reviewer again for his/her time and positive feedback on the paper. The revised paper is improved a lot from the theoretical and experimental aspects. We hope that the reviewer will be satisfied with the responses and the supplemented results as well, and then consider revising the assessment in support of the revised paper. We may make further improvements according to your feedback.

---

> > > ### Author Response · Authors · 2025-11-24
> > > **Update of the Response to the Reviewer qnFP**
> > >
> > > According to the discussion results to other reviewers, we have revised the manuscript and provide the following revision:
> > >
> > > **1.** We replace the result in Fig. 3 of the LyaGuide with a larger hyperparameter $\delta$, such that the revised figure show significant  improvement of the guidance compared to the original methods.
> > >
> > > **2.** We rephrase the presentation in Theorem 3.2, to explain the equivalence in our theory more clearly as
> > >
> > > ```
> > > There exists a control that is both locally Lyapunov stable and generates the guided probability path, which therefore accelerates the sampling process of guided flows.
> > > ```
> > > From this point, the pseudo-projection in Theorem 4.1 is designed solely to enforce the Lyapunov decrease condition, and, in general, it does not preserve the weighted continuity equation (Eq. 10 in the appendix). In this sense, our construction is partially heuristic: the aim is to obtain a guidance term that is provably Lyapunov-stable, thereby accelerating sampling and improving robustness, even though the divergence constraint may no longer be exactly satisfied after projection. The acceleration and robustness effects are reflected in Fig. 5 and Table 2 in the revised manuscript.
> > >
> > > Providing a rigorous correction operator that simultaneously guarantees both local Lyapunov stability and the weighted continuity equation remains an interesting and nontrivial problem. In this work, we focus on the advantages coming from the Lyapunov stability, so we leave the problem for a future direction. We have added a discussion of this limitation and its implications in the revised manuscript; please refer to Appendix A.2.6 for details.

---

> > > > ### Comment · Reviewer_qnFP · 2025-11-26
> > > > **Official Comment by Reviewer qnFP**
> > > >
> > > > I want to thank the authors for this exciting paper and for their patient responses. I have decided to raise my score. This is a good paper, and I wish you the best of luck with your publication.
> > > >
> > > > But Line 185 also has a typo "exists a a continuously ".

---

> > > > > ### Author Response · Authors · 2025-11-26
> > > > > **Response to the Reviewer qnFP**
> > > > >
> > > > > We sincerely thank the reviewer for the very positive feedback and for taking the time to re-evaluate our work. We truly appreciate your encouraging comments and are grateful that you decided to raise your score.
> > > > >
> > > > > Thank you also for catching the typo on Line 185 (“exists a a continuously”). We have corrected it in the revised manuscript.
> > > > >
> > > > > We are very grateful for your constructive comments and support.

---

### Official Review · Reviewer_mrD2 · 2025-11-01

**Soundness:** 3
**Presentation:** 3
**Contribution:** 2
**Rating:** 4
**Confidence:** 3

**Summary:**

The paper introduces LyaGuide, a novel framework that unifies various post-training guidance methods for flow-matching models using Lyapunov control theory. The central idea is to view the guidance process in generative modeling as a stabilization problem: the guidance term acts as a control input ensuring the system’s convergence toward a desired distribution.

**Strengths:**

1. The theoretical framework is sound and general, unifying several different scenarios of guidance in flow-based model.

**Weaknesses:**

1, Experiments are neither sufficient. Both reward-guided scenario (scenario 1) and prior-knowledge one (scenario 2) are equipped with highly complete empirical benchmarks, for example, GenEval for scenario 1 and normal image generation for scenario 2. The effectiveness of the guidance method should be evaluated on this well-established benchmarks.

2, Baselines are missing. Since both RLHF and prior-knowledge guidance are included in the theoretical framework, baselines for these two tasks should be then considered, like Flow-GRPO for RLHF and AutoGuidance (it also works on flow matching models empirically).

**Questions:**

See Weakness.

---

> ### Author Response · Authors · 2025-11-26
> **Response to the Reviewer mrD2**
>
> We thank the reviewer for raising the important concerns regarding benchmark sufficiency and baseline completeness. In the revised manuscript, we have substantially expanded our empirical evaluation along both the RL-guided and EBM-guided scenarios.
>
> **Regarding RLHF-style guidance baselines:** Although Flow-GRPO is suggested as a potential baseline, its GitHub official implementation only provides training scripts without pretrained checkpoints or test-time inference code. Reproducing Flow-GRPO requires training large text-to-image flow models from scratch, a process reported to take thousands of GPU hours, which is beyond the computational budget of our revision cycle. For this reason, Flow-GRPO cannot be meaningfully included as a reproducible baseline. To address this concern, we instead adopt the RL-guided planners introduced in the ICML 2025 work on controlled flow matching [R1], which provides fully open-sourced sampling and evaluation pipelines. This allows us to perform a fair and reproducible comparison. As shown in Table 2 in the revision, LyaGuide improves the normalized return across nearly all D4RL locomotion tasks, consistently outperforming the corresponding baseline guidance methods under both CFM and OT-CFM settings.
>
> **Regarding prior-knowledge or energy-based guidance:** To further strengthen our empirical coverage, we additionally include experiments on EBM sampling. In particular, we integrate LyaGuide into the NeurIPS 2025 Energy Matching framework [R2], which is a state-of-the-art EBM method with publicly released code and pretrained models. The new experiments (reported in Appendix A.5.3) demonstrate that LyaGuide accelerates the convergence of Langevin dynamics, reduces the number of required integration steps, and produces samples closer to the target distribution under the same computational budget. These results empirically confirm our theoretical claim that LyaGuide enhances the contractive structure of energy-driven vector fields.
>
> Overall, the newly added RL and EBM experiments significantly broaden the empirical support of our framework and show that LyaGuide provides consistent and reproducible improvements across multiple well-established benchmark settings.
>
> Finally, we would like to thank the reviewer again for the constructive feedback on the paper. The revised paper is improved a lot from the experimental aspects. We hope that the reviewer will be satisfied with the responses and the supplemented results as well, and then consider revising the assessment in support of the revised paper. We may make further improvements according to your feedback.
>
> References
>
> [R1] Feng, R., Yu, C., Deng, W., Hu, P. &amp; Wu, T.. (2025). On the Guidance of Flow Matching. Proceedings of the 42nd International Conference on Machine Learning, in Proceedings of Machine Learning Research 267:16993-17029 Available from https://proceedings.mlr.press/v267/feng25s.html.
>
> [R2] Balcerak, M., Amiranashvili, T., Terpin, A., Shit, S., Bogensperger, L., Kaltenbach, S., ... & Menze, B. (2025). Energy Matching: Unifying Flow Matching and Energy-Based Models for Generative Modeling. arXiv preprint arXiv:2504.10612.

---

> > ### Author Response · Authors · 2025-11-26
> > **Update of the Response to the Reviewer mrD2**
> >
> > According to the suggestions of the other reviewers, we also supplemented the experiments from other perspectives:
> >
> > **How does the sampling step of flow matching influence the performance of LyaGuide?**
> >
> > To investigate how the sampling time horizon and the number of flow-matching steps affect LyaGuide, we conducted an early-inference-termination experiment. Specifically, instead of integrating the flow matching dynamics up to $t=1$, we stop the ODE at intermediate times and measure the Wasserstein-2 distance between the partially evolved samples and the target distribution. The results are shown in Figure 5. Across all eight methods, LyaGuide consistently achieves lower W-2 distances at every inference time, indicating that its trajectories contract toward the target distribution significantly faster than those of the original guidance fields. In particular, the gap between the baseline and LyaGuide is already substantial at early inference times (e.g., $t \approx 0.4$), demonstrating that the Lyapunov-based correction accelerates convergence even when the flow is only partially executed.
> >
> > **How sensitive is the method to scaling parameters $(\delta,k)$ in the projection and control terms?**
> >
> > We systematically investigate the influence of the parameters $(\delta,k)$ in an ablation study on synthetic dataset. The results are summarised in Table 2 and Figure 10. Based on these new results, we address the reviewer's concern as follows:
> >
> > ```Sensitivity with respect to $\delta$.``` The parameter $\delta$ appears both in the Lyapunov convergence rate in Prop 3.1 and in the explicit projection term in Theorem 4.1. From Figure 10, we observe that:
> > - Overall, the method is robust to $\delta$ across a broad range of values.
> > - As $\delta$ increases, the generated samples concentrate more strongly near the minima of the Lyapunov function V.
> > - This stronger contraction improves convergence but also reduces the probability of visiting low-density regions, thereby decreasing exploration capability.
> > In other words, large $\delta$ induces over-contraction, which may excessively bias sampling towards high-density areas.
> > Therefore, $\delta$ can be tuned according to the task requirements. The results also show excessively large $\delta$ may lead LyaGuide to underperform compared to the original method. Based on our empirical findings, a practical recommendation is to choose $\delta\in(0,1]$.
> >
> > ```Sensitivity with respect to $k$ (gradient-based guidance only)```.
> > The control coefficient $k$ only affects gradient-based guidance methods (e.g., energy or score-based guidance), appears in the initial guidance term $-k\nabla V$ without LyaGuide. So we evaluate $k$ exclusively on the gradient-based variants (Table 2).
> > The results show that:
> > - Increasing $k$ generally improves sample quality, as stronger control in the Lyapunov direction facilitates more effective contraction of the gradient flow.
> > - The benefit of larger $k$ persists across different $\delta$ values, although extreme $\delta$ can diminish this improvement.
> > This confirms that $k$ acts as a refinement mechanism for stabilising gradient-based methods, rather than a global sensitivity parameter.
> >
> > These results further show that LyaGuide reduces the dependence of flow matching on fine integration granularity: even with fewer integration steps or shorter time horizons, the guided dynamics maintain stability and approach the target distribution more rapidly than the unguided flows. This suggests that the Lyapunov structure not only improves final-time sample quality but also enhances transient convergence efficiency. The results are also consistent with theoretical intuition, since we know that the Lyapunov control steers the controlled flow towards the minimizers of Lyapunov function with an exponential convergence rate, so it naturally accelerates the sampling process of flow matching.
> >
> > **Brief Summary**: Both of the supplemented results demonstrate the advantages our LyaGuide over the existing guidance methods in terms of the quality improvement, acceleration and robustness. And our work is the first to propose and validate the unifying theoretical framework that equates the guidance generative models with Lyapunov control problems, which paves a new direction for the subsequent research.

---

### Author Response · Authors · 2025-11-29
**Response to Area Chair**

We would like to thank the AC and reviewers for your time, efforts, and valuable comments and suggestions, which have helped us significantly improve the quality of this work in several directions. Accordingly, we have made substantial revisions to both the theoretical exposition and the empirical evaluation. The revised article now contains clearer analytical results, extended discussions of the proposed LyaGuide framework, and several additional experiments on comparison studies. Specifically, our main changes in the revision include:

**Theoretical clarification and extensions.**

 We clarified the equivalence between guidance-compatible controls and Lyapunov-stable controls, and reorganised the proof of Theorem A.2.1 to make the “deformation” between the two more explicit. We emphasised that all our theory provide guarantees for both local and global stability for the relevant tasks, depending whether the task-relevant minimas are in the multi-modal case or single-modal case. We also clarified how Lyapunov’s positive definiteness and negative-derivative conditions are handled in practice (via affine transformations of $V$ and the pseudo-projection in Theorem 4.1), and discussed regions of attraction and their boundaries in response to the stability-related concerns.

**New experiments on convergence, ablations, RL guidance, and EBMs.**

(1) To address the request for stronger benchmarks and RL-related baselines, we incorporated LyaGuide into standard offline RL planning on the D4RL Locomotion tasks, using the open-sourced RL-guided flow matching framework from ICML~2025. Under both CFM and OT-CFM, LyaGuide consistently improves the normalized return over the underlying guidance methods.

(2) In addition, we added EBM experiments by integrating LyaGuide into the ICML2025 Energy Matching framework. The new results (reported in the appendix) show that LyaGuide accelerates Langevin dynamics, reduces the required integration steps, and produces samples closer to the target distribution under the same computational budget.

(3) We added early-inference-termination experiments on synthetic data to quantify how LyaGuide affects convergence speed: when the flow is truncated before $t=1$, LyaGuide consistently achieves lower W-2 distance at every inference time, demonstrating substantially faster contraction toward the target distribution.

(4) We further performed an ablation study on the scaling parameters $(\delta,k)$, showing that the method is robust to $\delta$ in a broad range (with a practical recommendation $\delta\in[0,1]$) and that $k$ mainly refines gradient-based guidance.

All the supplemented numerical studies show the superiority of our LyaGuide over the existing guidance approaches, and demonstrate the efficacy of LyaGuide in synthetic dataset, images generation task, EBMs, and RL tasks.

**Improved presentation and corrections.**

We revised the introduction and Section 3,4 to give a more intuitive explanation of why the control term $c$ is “derived from $V$”, why stabilising controllers are central in cybernetics-style problems, and how $g_\phi$ is defined and used. We refined the caption and legend of Figure 2, and improved the overall readability of the main text and appendix to avoid possible misinterpretations (e.g., about “rigorous stability guarantees” and global vs. local stability).

**Summary of reviewers' replies.**

During the rebuttal period, Reviewers wGTM, qnFP, and NCg4 have expressed appreciation for our revisions and have accordingly raised their scores to 6. Reviewer mrD2 has not yet responded to the revision, but we have carefully addressed all points raised by this reviewer in the updated manuscript.



Finally, we thank all the reviewers again for their insightful comments, and we sincerely thank the AC for the time and efforts to review our work. We believe that the revised article is much improved not only from the theoretical aspect but also from the experimental aspect. We would greatly appreciate it if you and the reviewers could recognize the additional contributions and our efforts in this revision.

---

### Meta-Review · Area_Chair_cL6a · 2026-01-08

**Summary:**

This paper introduces LyaGuide, a unifying framework that formulates post-training guidance for flow matching models through the lens of Lyapunov control theory. By interpreting energy or reward functions as Lyapunov functions and guidance terms as stabilizing controls, the method provides a principled approach to improving the stability and convergence of guided flows. A key technical contribution is a closed-form pseudo-projection operator that enforces a Lyapunov decrease condition while remaining lightweight and compatible with existing guidance methods. Experiments on synthetic benchmarks and image inverse problems demonstrate improved stability, faster transient convergence, and modest gains in final quality.

**Reviewer Concerns:**

Across the discussion, the authors substantially improved both clarity and empirical support. Several reviewers’ concerns regarding over-claiming global stability were resolved by clearly restricting the theory to local Lyapunov stability within regions of attraction, which is appropriate for multimodal generative settings. Questions about the equivalence between guidance and Lyapunov stability were addressed with clearer explanations and revised statements, and multiple presentation issues (definitions, figures, notation) were corrected. Empirically, the authors added early-termination analyses, Wasserstein-distance trajectories, and parameter sensitivity studies, demonstrating that LyaGuide consistently improves transient convergence and robustness, not just final-time metrics. Multiple reviewers explicitly indicated increased confidence and raised their scores after these clarifications and additions.

Despite these improvements, the overall scope and impact of the contribution remain limited relative to the strength of the batch. The method primarily acts as a stabilizing wrapper around existing guidance techniques rather than enabling fundamentally new capabilities, and the final quantitative gains on realistic tasks are often modest. While the Lyapunov framing is elegant and unifying, its practical benefit depends on the availability or learnability of suitable Lyapunov functions. In addition, the authors acknowledge that the pseudo-projection step is partially heuristic, as it may break exact continuity constraints, with a fully principled correction left for future work. Given the number of strong acceptances already in the batch, these factors make it difficult to prioritize this paper over submissions with broader empirical impact or more substantial performance gains.

**Reviewer Scores:**

Reviewer qnFP and Reviewer wGTM have claimed to raise their scores. Other reviewers are likely to maintain their scores.

---

### Decision · Program_Chairs · 2026-01-26

Reject